# Decision Tree Induction Through LLMs via Semantically-Aware Evolution

**Tennison Liu**[*], **Nicolas Huynh**[*] **& Mihaela van der Schaar**
DAMTP, University of Cambridge
Cambridge, UK
`{tl522,nvth2,mv472}@cam.ac.uk`

## Abstract

Decision trees are a crucial class of models offering robust predictive performance and inherent interpretability across various domains, including healthcare, finance, and logistics. However, current tree induction methods often face limitations such as suboptimal solutions from greedy methods or prohibitive computational costs and limited applicability of exact optimization approaches. To address these challenges, we propose an evolutionary optimization method for decision tree induction based on genetic programming (GP). Our key innovation is the integration of semantic priors and domain-specific knowledge about the search space into the optimization algorithm. To this end, we introduce `LLEGO`, a framework that incorporates semantic priors into genetic search operators through the use of Large Language Models (LLMs), thereby enhancing search efficiency and targeting regions of the search space that yield decision trees with superior generalization performance. This is operationalized through novel genetic operators that work with structured natural language prompts, effectively utilizing LLMs as conditional generative models and sources of semantic knowledge. Specifically, we introduce *fitness-guided* crossover to exploit high-performing regions, and *diversity-guided* mutation for efficient global exploration of the search space. These operators are controlled by corresponding hyperparameters that enable a more nuanced balance between exploration and exploitation across the search space. Empirically, we demonstrate across various benchmarks that `LLEGO` evolves superior-performing trees compared to existing tree induction methods, and exhibits significantly more efficient search performance compared to conventional GP approaches.

## 1 Introduction

Decision trees are fundamental models, which are widely utilized across various domains, including finance, healthcare, and bioinformatics (Morgan & Sonquist, 1963; Che et al., 2011; Soleimanian et al., 2012). These hierarchical models recursively partition the feature space, creating a tree-like structure where internal nodes represent decision rules based on feature values, and leaf nodes correspond to class labels or predicted values. Decision trees are particularly appealing as they offer both predictive accuracy and interpretability, which have stood the test of time against recently developed black-box predictive models (Borisov et al., 2022; Grinsztajn et al., 2022).

However, decision tree induction represents a challenging optimization problem. Finding the optimal tree given a training dataset is NP-complete (Laurent & Rivest, 1976), often necessitating the use of heuristic algorithms (Quinlan, 1986). While computationally efficient, these heuristics yield approximate, locally greedy solutions that sacrifice some degree of performance and global optimality (Rokach & Maimon, 2005). Exact optimization methods have been developed to address these limitations, but they face their own constraints. Their computational complexity typically scales exponentially with problem size, limiting their applicability to restricted search spaces, and specific problem types (e.g., binary classification) (Verwer & Zhang, 2019; Lin et al., 2020).

Genetic programming (GP) is a class of evolutionary algorithms which offers a promising middle ground for decision tree induction, balancing computational efficiency with global optimization

---

[*]Equal contributions.

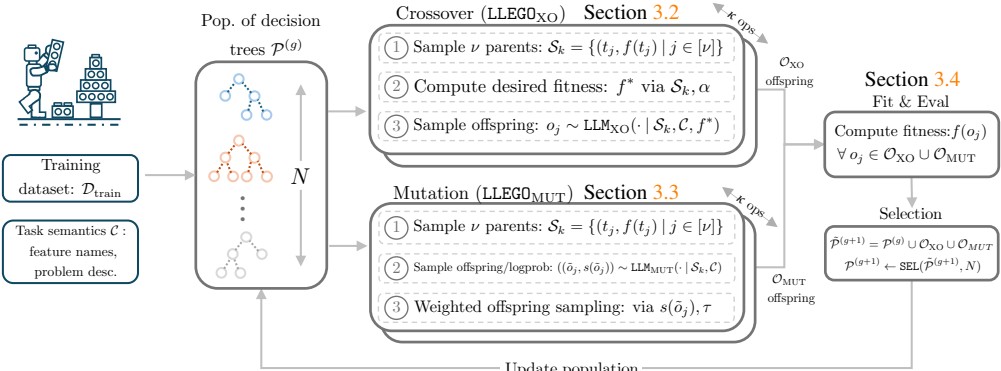

Figure 1: **LLEGO Overview.** In each generation $g \in [G]$, a population of trees $\mathcal{P}^{(g)}$ is evolved through **crossover** $\mathcal{O}_{\text{MUT}} = \text{LLEGO}_{\text{XO}}(\mathcal{P}^{(g)}, \mathcal{C}; \alpha)$ and **mutation** $\mathcal{O}_{\text{MUT}} = \text{LLEGO}_{\text{MUT}}(\mathcal{P}^{(g)}, \mathcal{C}; \tau)$. Subsequently, the offsprings $\mathcal{O}_{\text{XO}} \cup \mathcal{O}_{\text{MUT}}$ are **evaluated** for fitness on $\mathcal{D}_{\text{train}}$; and **selection** preserves the top-$N$ trees, $\mathcal{P}^{(g+1)} \leftarrow \text{SEL}(\tilde{\mathcal{P}}^{(g+1)}, N)$, where $\tilde{\mathcal{P}}^{(g)} = \mathcal{P}^{(g)} \cup \mathcal{O}_{\text{XO}} \cup \mathcal{O}_{\text{MUT}}$.

of the tree structure (Koza, 1994a;b). Inspired by principles of evolution, GP algorithms evolve a population of candidate solutions through iterative application of genetic operators such as *crossover* and *mutation*. They are particularly well-suited for optimizing combinatorial problems with discrete, variable-length search spaces, as is the case in decision tree induction (Koza, 1990; Tanigawa & Zhao, 2000; Kuo et al., 2007; Lahovnik, 2024). While much research in GP has focused on designing genetic operators to enhance search efficiency, these approaches face inherent limitations that constrain their exploratory effectiveness. Key challenges include the difficulty of incorporating semantic information into genetic operations—resulting in primarily structural, unguided mechanisms—and the narrow operational contexts that limit global exploration.

**Key considerations.** The key insight of this work is to employ *large language models* (LLMs) to design semantically-aware genetic operators for decision tree induction. LLMs are powerful generative models capable of learning distributions over discrete and variable-length sequences given only few-shot examples (Radford et al., 2019; Brown et al., 2020). We utilize LLMs as the foundation for crossover and mutation operators, leveraging their encoded semantic priors to create meaningful distributions over potential offspring. Building on LLMs, our approach introduces *fitness-guided* crossover and *diversity-guided* mutation operators within a GP framework, enabling efficient search space exploration that contrasts with the structural focus and unguided nature of conventional genetic operators. By representing decision trees in natural language, we also enable *higher-arity* genetic operations, capable of operating over multiple trees simultaneously.

**Contributions.** *Conceptually*, we propose a novel GP algorithm that leverages semantic priors contained in LLMs to enhance search efficiency and performance on challenging decision tree induction problems. *Technically*, we introduce LLEGO (LLM-Enhanced Genetic Operators), which uses LLMs to define two key search operators: *fitness*-guided crossover that steers the search towards promising regions using a target fitness; and *diversity*-guided mutation that employs log-probabilities to evolve solutions in under-explored search regions. *Empirically*, on a wide range of classification and regression tabular benchmarks, we demonstrate that LLEGO significantly improves search efficiency and consistently evolves trees with superior generalization performance.

## 2 PRELIMINARIES

### 2.1 DECISION TREE INDUCTION

Decision tree induction is the problem of learning a decision tree $t \in \mathcal{T}$ from a training dataset $\mathcal{D}_{\text{train}} = \{(x_i, y_i)\}_{i=1}^n$, where $x_i \in \mathcal{X} \subseteq \mathbb{R}^d$ denotes a $d$-dimensional input and $y_i \in \mathcal{Y}$ denotes the output. Decision trees recursively partition the input space $\mathcal{X}$ into hierarchical, disjoint regions. In this work, we focus on binary decision trees, where splits partition regions in two subregions. These regions define a set of leaf nodes $R = \{R_1, R_2, ..., R_L\}$, where each leaf $R_l$ is assigned a constant $c_l$ (Hastie et al., 2009). This in turn yields a predictor $t : \mathcal{X} \to \mathcal{Y}$ which is defined by $t(x) = \sum_{l=1}^L c_l I(x \in R_l)$, where $I(\cdot)$ is the indicator function. Constructing decision trees from a

training dataset is a challenging optimization problem, since full tree optimization has been proven to be an *NP*-complete problem (Laurent & Rivest, 1976). Greedy algorithms like CART (Breiman et al., 1984) build trees top-down, offering computational efficiency but sacrifices performance by only finding locally optimal trees. By comparison, exact optimization methods (Lin et al., 2020) provide theoretical guarantees of global optimality, but scale exponentially with problem size, and apply only to classification tasks and objective functions of specific forms.

## 2.2 GENETIC PROGRAMMING

Genetic Programming (GP) is a class of evolutionary algorithms for searching combinatorial spaces and offers a flexible middle ground between greedy and exact optimization methods. The fundamental objective of GP is to evolve trees $t \in \mathcal{T}$ to maximize a fitness function $f : \mathcal{T} \to \mathbb{R}$, where $\mathcal{T}$ is the combinatorial search space (Koza, 1994a). In GP, each individual is described by the tuple $(t, f(t))$ containing a tree and its fitness. We denote this population of $N$ individuals $\mathcal{P} = \{(t_1, f(t_1)), (t_2, f(t_2)), \ldots, (t_N, f(t_N))\}$, with $N \in \mathbb{N}$. The algorithm evolves the population across $G \in \mathbb{N}$ generations. In each generation $g \in [G]$, the population $\mathcal{P}^{(g)}$ undergoes three key genetic operations: selection and two *variation* operators (crossover and mutation).

**Selection.** The selection mechanism preserves performant trees across generations, placing *selection pressure* on sufficient exploitation and ensures convergence (Goldberg, 1989). The $N$-ary selection operator is defined as $\text{SEL} : \mathcal{T}^N \times \mathbb{R}^N \to \Delta(\mathcal{T}^N)$, where $\Delta(\mathcal{T}^N)$ represents the probability simplex over $\mathcal{T}^N$. Often, selection operators implicitly create this probability distribution over $\mathcal{P}$, wherein trees with higher fitness are more likely to be preserved.

**Crossover.** The crossover operator combines the genetic material of multiple candidate trees to generate performant offspring (Langdon & Poli, 2013). Crossover is an $\nu$-ary operator, denoted $\text{XO} : \mathcal{T}^\nu \to \Delta(\mathcal{T})$, taking in $\nu$ parents to generate an offspring $o \in \mathcal{T}$, sampled as, $o \sim p_{\text{XO}}(\cdot \mid \mathcal{S})$, where $\mathcal{S}$ is usually a pair of parents ($\nu = 2$) sampled uniformly from the population. $\text{XO}$ induces the offspring distribution $p_{\text{XO}}$, which can be interpreted as a uniform distribution over all trees producible by the crossover operator. For example, a popular version of $\text{XO}$ is subtree crossover, where randomly selected subtrees from two parent trees are swapped (Koza, 1994a).

**Mutation.** The mutation operator promotes global exploration, thus mitigating premature convergence to local optima (Goldberg, 1989). An $\nu$-ary mutation operator $\text{MUT} : \mathcal{T}^\nu \to \Delta(\mathcal{T})$ performs random modifications to parents to generate an offspring $o \sim p_{\text{MUT}}(\cdot \mid \mathcal{S})$. Traditionally, mutation operates on a single parent tree ($\nu = 1$) and $p_{\text{MUT}}$ is uniform over the set of trees that can be generated through a mutation operator (e.g., random insertion or replacement of nodes).

## 2.3 DESIDERATA

We can conceptualize a variation operator, $v$, as implicitly defining a sampling distribution $p_v(o \mid \mathcal{S})$ that depends on the parent trees $\mathcal{S}$ and its rules. More formally, $p_v(o \mid \mathcal{S}) = \int p_v(o \mid \mathcal{S}, g) p_v(g \mid \mathcal{S}) \, dg$, where $p_v(g \mid \mathcal{S})$ represents the prior distribution of applying a specific genetic operation (e.g., pairs of nodes when $v$ corresponds to subtree crossover), and $p_v(o \mid \mathcal{S}, g)$ is the likelihood of an offspring given parents and genetic operation (generally, 1 if producible, and 0 otherwise). As such, genetic operators can be viewed as sampling from a posterior distribution over offspring, where the prior is encoded through the operator's design. While these variation mechanisms are core to GP, they present notable limitations that negatively impact search performance, leading to the following desiderata:

- **Semantic priors**: Conventional variation operators encode inductive biases on *structural* properties, crucially lacking any considerations for tree semantics, i.e., $p_v(g \mid \mathcal{S})$ is independent of the semantics. This can be problematic, as small changes to the structure can lead to disruptive changes to the functional behavior (an issue known as rough genotype-phenotype mapping Rothlauf et al. (2011)). This can be improved through integration of problem semantics into the prior $p_v(g \mid \mathcal{S}, \mathcal{C})$, where $\mathcal{C}$ represents the semantics, leading to the sampling distribution $p_v(o \mid \mathcal{S}, \mathcal{C})$.
- **Guided variations**: Generally, conventional variation mechanisms place an uninformative distribution over any genetic operations. For example, they might consider any structural operations as equally likely, i.e., $p(g \mid \mathcal{S})$ is uniform. This lack of search guidance can lead to inefficient exploration and slower convergence, which can be improved with more informative priors that

    prioritize offsprings that are more semantically meaningful, or likely to improve fitness and cover unexplored regions of the search space.

- **Broader context**: The designs of existing operators often constrain the arity of allowed operations (e.g., it is difficult to define valid operators on $> 2$ trees), restricting offsprings to local exploration. In contrast, including more parents in genetic operations can improve global exploration.

A line of work has aimed to address some of these desiderata. Notably, previous works in semantic GP have attempted to address the first two limitations with variation operators, which consider the semantics of solutions (Krawiec & Lichocki, 2009; Moraglio et al., 2012; Krawiec & Pawlak, 2013). In the semantic GP literature, a solution's semantics typically refers to the *functional output* of a solution, i.e., $h(t) = [t(x_1), t(x_2), \ldots, t(x_n)] \in \mathbb{R}^n$. In contrast, our work uses the term to describe *domain knowledge* about the solution space encoded in the LLM. Additionally, semantic GP is limited to application-specific definitions of semantics that restrict its broader applicability. Crucially, no comparable semantically-aware method has been developed for decision tree induction.

## 3 LLEGO: GENETIC OPERATORS WITH SEMANTIC PRIORS

Designing genetic operators that satisfy the aforementioned desiderata using conventional methods has proven difficult. In this work, we build on the insight that LLMs are powerful generative models that can be employed as semantically aware variation operators. Indeed, LLMs possess several properties that make them appealing (Meyerson et al., 2023; Lehman et al., 2023). Firstly, we utilize LLMs as a source of *semantic prior* $p_{\text{LLM}}(g \,|\, \mathcal{S}, \mathcal{C})$, as they contain rich semantic knowledge of the problem and tree solutions, forming the basis of variation operators (Xie et al., 2021). Secondly, they are able to reason over and learn from in-context examples to identify high-potential patterns in candidate trees and produce *guided variations* towards desired regions. Lastly, their relatively large context window facilitates utilization of *broader context*, increasing arity of feasible genetic operations. Building on this semantic prior, we design genetic operators through mechanisms that incorporate fitness information and log probabilities of generated offspring, to further improve exploration and exploitation abilities.

**Method overview.** At a high level, LLEGO represents trees and frames genetic operations in natural language. Specifically, each genetic operation is realized through a distinct prompt which contains parent trees, semantics, and auxiliary information. We introduce *fitness*-guided crossover $\text{LLEGO}_{\text{XO}}$ that performs in-context learning over solutions and their fitness, and generates offspring conditioned on a desired fitness $f^*$, to steer evolution towards high-performing regions. Additionally, we propose *diversity*-guided mutation $\text{LLEGO}_{\text{MUT}}$, which uses the log probabilities of candidate offspring to construct a weighted sampling distribution, prioritizing efficient exploration that cover unexplored search regions. We note that the level of fitness- or diversity-guidance is intentionally controllable through two hyperparameters, $\alpha$ and $\tau$ that correspond to different degrees of exploitation *vs.* exploration. An overview of our method is visualized in Figure 1.

### 3.1 LLEGO PROMPT DESIGN

The genetic operations are performed through natural language queries to the LLM. While the specific structure of each query differs, they are constructed from three essential components. For an extended description of prompts and examples, please refer to Appendix B.

1. **Task context.** Denoted as $\mathcal{C}$ and encapsulates the semantic description of the problem, including semantic and statistical descriptions of the input space features $\mathcal{X}$, output label $\mathcal{Y}$, and characteristics of the overall dataset, e.g., number of samples or features.
2. **Parent solutions.** This contains the solution representation and fitness of each parent, which are serialized into natural language and provided as few-shot examples in each genetic operation.
3. **Task-specific instructions.** For each genetic operator, we include task-specific instructions on offspring generation and guidelines on the format of the response.

### 3.2 *Fitness*-GUIDED CROSSOVER OPERATOR

Traditional crossover operators are not *semantically aware*, as they randomly select subtrees from parents to be recombined into an offspring. This ignores patterns in the parents, introducing the

possibility for performant substructures to be destroyed through random perturbations. Additionally, they do not make use of parent fitness explicitly to guide offspring generation (i.e., no *fitness guidance*), foregoing any informative signals on correlations between fitness and solution structure. We seek to address these factors in our fitness-guided crossover operator. More specifically, the crossover operation includes three steps: *(1)* sampling a subset of parents, weighted by parent fitness, *(2)* compute desired fitness $f^*$ based on parent fitness, *(3)* constructing the prompt and querying the LLM to generate offsprings, conditioning on $f^*$ (see Figure 2).

**Parent sampling.** Each crossover operation is conditioned on parents, which are sampled from the current population. We utilize a roulette-wheel mechanism (Blickle & Thiele, 1996) to favour existing solutions with high fitness. Specifically, we aim to sample a set of $\nu \in \mathbb{N}$ parents for each crossover operation, where the sampling weights $\theta = (\theta_1, \ldots, \theta_N)$ are proportional to the solutions' fitness. These weights define a categorical distribution $\text{Cat}_N(\theta)$, based on which we sample parents without replacement. Intuitively, solutions with higher fitness are more likely to be involved in genetic operations. We use $\mathcal{S}_k$ to represent the set of parents sampled for operation $k \in [\kappa]$, where $\kappa \in \mathbb{N}$ is the number of crossover operations performed.

**Crossover through fitness guidance.** To perform crossover, we utilize both the tree structures $t_i$ and the fitness metric $f(t_i)$ to create few-shot prompts. For each of the sampled parents in $\mathcal{S}$, we serialize the tree into natural language as a nested dictionary, which we denote as $t_i^{\text{nl}}$, where each intermediary key represents the splitting condition (feature name and splitting value) and the leaf item represents the value of the leaf node. Please refer to Figure 10 for more description of this representation. Each example is then constructed as *"fitness: $f(t_i)$, tree: $t_i^{nl}$"* and we use $\mathcal{S}^{\text{nl}}$ to represent the serialized few-shot prompt. We further condition the generation by specifying a *desired fitness* $f^*$ in the prompt to steer the generation towards high-fitness re-

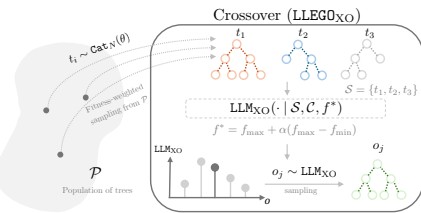

Figure 2: **LLEGO$_{\text{XO}}$.** In each operation, the crossover operator *(1)* samples a set of parents $\mathcal{S}$ weighted by their fitness, *(2)* computes the desired fitness $f^*$ using $\mathcal{S}$ and $\alpha$, and *(3)* samples offspring via the LLM.

gions. This steering is controlled by a hyperparameter $\alpha$, where $f^* = f_{\max} + \alpha(f_{\max} - f_{\min})$, with $f_{\max}$ and $f_{\min}$ the best and worst fitness in $\mathcal{S}$ respectively. Intuitively, $f^*$ is defined relative to the best parent fitness, with the improvement proportional to the observed variability. A positive $\alpha$ defines $f^*$ to improve over the best fitness in the parent set, whereas $-1 \leq \alpha < 0$ results in a more conservative target fitness that is within the observed fitness range.

We generate offsprings as $o_j \sim \text{LLM}_{\text{XO}}(\cdot \mid \mathcal{S}^{\text{nl}}, \mathcal{C}, f^*)$, by sampling from an LLM conditioned on the parents $\mathcal{S}^{\text{nl}}$, the task context $\mathcal{C}$, and target fitness $f^*$ controlled by $\alpha$. We write the complete crossover operation as $\mathcal{O}_{\text{XO},k} = \text{LLEGO}_{\text{XO}}(\mathcal{P}^{(g)}, \mathcal{C}; \alpha)$, where $\mathcal{O}_{\text{XO},k}$ is the set of offspring generated from the operation $k \in [\kappa]$. $\alpha$ controls the level of extrapolation, and we systematically investigate its effect in Section 5.2. By framing crossover using natural language, our crossover operator naturally allows for an arity $\nu$ strictly than 2, by including additional parents as in-context examples through $\mathcal{S}^{\text{nl}}$.

### 3.3 *Diversity*-GUIDED MUTATION OPERATOR

On the other side of the coin is the mutation operator, where the objective is to efficiently traverse under-explored areas in the search space to improve diversity and escape local minima. Traditional mutation operators can be viewed as inducing a uniform distribution over the space of solutions one random mutation away from the parent. However, this does not consider whether such mutations are *semantically meaningful*. To contextualize this, imagine the space one mutation away from a decision tree; many of these mutations are highly unlikely to be interesting or optimal given some degree of domain knowledge, resulting in inefficient exploration. In this setting, our mutation operator uses its semantic prior to effectively guide exploration, enabling more efficient *diversity*-driven exploration.

**Parent sampling.** As before, each mutation operation is conditioned on a set $\mathcal{S}$ of $\nu$ parents. However, whereas for crossover, parents are sampled based on their fitness, for mutation, parents are randomly sampled from the population to increase the diversity of $\mathcal{S}$. Specifically, $\mathcal{S} = \{(t_j, f(t_j)) \mid j \in [\nu], t_j \sim \text{Uniform}_N(\mathcal{P}^{(g)})\}$, where each solution has uniform probability of being selected as a parent. Future works should consider more advanced sampling schemes to improve parent diversity.

**Mutation with diversity guidance.** To perform mutation, we only include the tree structure $t_j$ to create few-shot prompts: each parent is serialized as *"tree: $t_j^{nl}$"*, to create $\mathcal{S}^{nl}$. Subsequently, we generate $\lambda'$ (where $\lambda' \geq \lambda$) *candidate* offsprings $\tilde{o}_j$, and track the negative log probabilities of the candidates obtained from the LLM, represented $s(\tilde{o}_j) = -p(\tilde{o}_j \mid \mathcal{S})$. Intuitively, $s(\tilde{o}_j)$ reflects the likelihood of the candidate offspring given the set of parents, with smaller values indicating that the candidate offspring has low probability under the current population distribution and hence that its inclusion can introduce more diversity at the population level. We provide further justification for the use of log probabilities in Section 5.2 and Appendix D.4, demonstrating their correlation with functional and structural distances between parent and offspring trees.

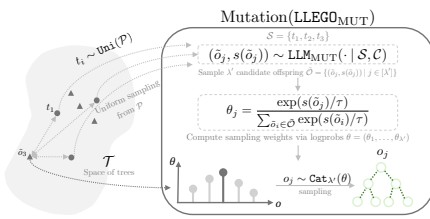

Figure 3: **LLEGO_MUT.** In each operation, the mutation operator: *(1)* samples a set of $\lambda'$ candidate offsprings $\tilde{\mathcal{O}}$, *(2)* computes sampling weights, $\theta$, inversely proportional to logprobs, with temperature $\tau$ controlling diversity, and *(3)* sample offspring via this weighted distribution $\texttt{Cat}_{\lambda'}(\theta)$.

As such, the candidate sampling step is represented as $(\tilde{o}_j, s(\tilde{o}_j)) \sim \texttt{LLM}_{\text{MUT}}(\cdot \mid \mathcal{S}^{nl}, \mathcal{C})$. Given this set of $\lambda'$ candidates, we select $\lambda$ offspring based on their log probabilities, i.e., $\mathcal{O}_{\text{MUT}} = \{(o_j, f(o_j) \mid j \in [\lambda], o_j \sim \texttt{Cat}_{\lambda'}(\theta)\}$, where $\theta = (\theta_1, \ldots, \theta_{\lambda'})$ and $\theta_j = \frac{\exp(s(\tilde{o}_j)/\tau)}{\sum_{i=1}^{\lambda'} \exp(s(\tilde{o}_j)/\tau)}$. Here, $\tau$ is the sampling temperature, where larger values of $\tau > 1$ results in a more uniform distribution over the candidate offspring, and lower values of $0 < \tau \leq 1$ would put more weight on candidates with lower likelihood. As such, we use $\tau$ and the log probabilities to guide the sampling of offspring with controllable levels of diversity. In Section 5.2, we empirically investigate the effect of $\tau$ on offspring diversity. In summary, we define the $k$-th mutation operation as $\mathcal{O}_{\text{MUT},k} = \texttt{LLEGO}_{\text{MUT}}(\mathcal{P}^{(g)}, \mathcal{C}; \tau)$.

### 3.4 END-TO-END ALGORITHM

Having detailed our LLM-based genetic operators, we now put together the end-to-end GP algorithm. The search is initialized with a set of $N$ solutions, $\mathcal{P}^{(0)}$. In each generation, we sample $N$ crossover offspring, represented as $\mathcal{O}_{\text{XO}}^{(g)}$, and mutation offsprings, represented as $\mathcal{O}_{\text{MUT}}^{(g)}$. This is performed through $\kappa$ genetic operations, with each operation involving $\nu$ parents, and generating $\lambda$ offsprings. The fitness of each solution is then calculated against the training set $\mathcal{D}_{\text{train}}$. For selection, we consider the set of candidates as the union of the solutions from the previous generation and the newly generated offsprings, i.e. $\tilde{\mathcal{P}}^{(g+1)} = \mathcal{P}^{(g)} \cup \mathcal{O}_{\text{XO}}^{(g)} \cup \mathcal{O}_{\text{MUT}}^{(g)}$. We use the *elitism* selection to select the top-$N$ unique solutions from the candidate population to preserve the highest quality solutions across generations (Goldberg, 1989). Here, top-$N$ is selected based on training set fitness. More formally, we denote this process as $\mathcal{P}^{(g+1)} \leftarrow \texttt{SEL}(\tilde{\mathcal{P}}^{(g+1)}; N)$. After $G$ generations of evolution, we select the solution with the highest *validation* fitness as the final solution.

## 4 RELATED WORKS

Our work relates to multiple strands of research, which we summarize in brief below. We provide an extended literature survey in Appendix A.1.

**Tree induction algorithms.** Existing algorithms for decision tree induction can be broadly categorized into three main classes: greedy, globally optimal, and GP algorithms. *Greedy* algorithms recursively construct a tree in a top-down approach, heuristically making locally optimal splits at each node (Breiman et al., 1984; Quinlan, 1986; 1993). While computationally efficient, these methods do not pursue global optimality. Recent works have proposed exact combinatorial methods to construct *optimal* decision trees (Verwer & Zhang, 2019; Hu et al., 2019; Lin et al., 2020; Aglin et al., 2020). These methods face two key limitations: exponential complexity scaling with tree depth and number of splits, and restricted applicability to specific objectives (primarily classification problems).

**Genetic programming.** *GP* approaches present a middle ground between search performance and computational efficiency (Koza, 1990; Tanigawa & Zhao, 2000; Lahovnik, 2024). GP builds on genetic operators that operate over structure (genotype) but can have disruptive effects because of the complex genotype-phenotype mapping (Rothlauf et al., 2011). This observation has motivated works

Table 1: **Performance on classification tasks.** Balanced accuracy ($\uparrow$) on 7 datasets, reporting mean$_{(std)}$ and emboldening best results. We also report average rank ($\downarrow$) for comparing baselines.

| Method | Breast | Compas | Credit | Diabetes | Heart | Liver | Vehicle | Rank ($\downarrow$) |
|---|---|---|---|---|---|---|---|---|
| | | | | *depth = 3* | | | | |
| CART | $0.941_{(0.009)}$ | $0.655_{(0.012)}$ | $0.668_{(0.021)}$ | $0.710_{(0.029)}$ | $0.734_{(0.068)}$ | $0.646_{(0.025)}$ | $0.903_{(0.021)}$ | $2.9_{(0.83)}$ |
| C4.5 | $0.938_{(0.012)}$ | $0.650_{(0.009)}$ | $0.579_{(0.030)}$ | $0.687_{(0.045)}$ | $0.704_{(0.019)}$ | $0.569_{(0.047)}$ | $0.857_{(0.039)}$ | $4.9_{(1.07)}$ |
| DL85 | $\mathbf{0.947_{(0.008)}}$ | $\mathbf{0.665_{(0.005)}}$ | $0.590_{(0.045)}$ | $0.703_{(0.027)}$ | $0.688_{(0.024)}$ | $0.598_{(0.034)}$ | $0.932_{(0.009)}$ | $3.0_{(1.73)}$ |
| GOSDT | $0.935_{(0.005)}$ | $0.641_{(0.004)}$ | $\mathbf{0.681_{(0.000)}}$ | $0.698_{(0.012)}$ | $0.651_{(0.086)}$ | $0.656_{(0.018)}$ | $0.852_{(0.047)}$ | $4.4_{(2.15)}$ |
| GATREE | $0.942_{(0.009)}$ | $0.647_{(0.005)}$ | $0.648_{(0.045)}$ | $0.681_{(0.027)}$ | $0.669_{(0.031)}$ | $0.626_{(0.037)}$ | $0.922_{(0.022)}$ | $4.3_{(1.11)}$ |
| LLEGO | $0.946_{(0.011)}$ | $0.652_{(0.004)}$ | $0.679_{(0.007)}$ | $\mathbf{0.713_{(0.015)}}$ | $\mathbf{0.736_{(0.024)}}$ | $\mathbf{0.672_{(0.019)}}$ | $\mathbf{0.937_{(0.017)}}$ | $\mathbf{1.6_{(0.79)}}$ |
| | | | | *depth = 4* | | | | |
| CART | $0.945_{(0.010)}$ | $0.660_{(0.011)}$ | $0.675_{(0.019)}$ | $0.704_{(0.026)}$ | $0.713_{(0.059)}$ | $0.632_{(0.063)}$ | $0.925_{(0.020)}$ | $2.9_{(0.90)}$ |
| C4.5 | $0.942_{(0.013)}$ | $0.660_{(0.006)}$ | $0.622_{(0.043)}$ | $0.699_{(0.024)}$ | $0.714_{(0.032)}$ | $0.585_{(0.046)}$ | $0.921_{(0.011)}$ | $3.6_{(1.13)}$ |
| DL85 | $0.939_{(0.012)}$ | $0.661_{(0.004)}$ | $0.576_{(0.017)}$ | $0.671_{(0.020)}$ | $0.706_{(0.058)}$ | $0.561_{(0.019)}$ | $0.898_{(0.065)}$ | $4.7_{(1.50)}$ |
| GOSDT | $0.938_{(0.007)}$ | $0.641_{(0.004)}$ | $0.680_{(0.002)}$ | $0.701_{(0.011)}$ | $0.677_{(0.028)}$ | $0.660_{(0.016)}$ | $0.885_{(0.019)}$ | $4.3_{(1.89)}$ |
| GATREE | $0.941_{(0.008)}$ | $0.650_{(0.007)}$ | $0.658_{(0.013)}$ | $0.675_{(0.038)}$ | $0.676_{(0.025)}$ | $0.633_{(0.047)}$ | $0.895_{(0.033)}$ | $4.6_{(0.98)}$ |
| LLEGO | $\mathbf{0.951_{(0.007)}}$ | $\mathbf{0.663_{(0.005)}}$ | $\mathbf{0.684_{(0.011)}}$ | $\mathbf{0.721_{(0.017)}}$ | $\mathbf{0.751_{(0.042)}}$ | $\mathbf{0.676_{(0.021)}}$ | $0.929_{(0.015)}$ | $\mathbf{1.0_{(0.00)}}$ |

on semantic GP (Krawiec & Lichocki, 2009; Moraglio et al., 2012; Krawiec & Pawlak, 2013), aiming to produce offspring that inherit semantics (phenotype) from parents. However, these approaches are highly domain-specific, and have not extended to tree induction, which is the focus of our work.

**LLMs and optimization.** Recent studies have explored LLMs for optimization tasks, with some works employing LLMs as variation operators (Meyerson et al., 2023). Examples of applications include code evolution (Lehman et al., 2023; Nasir et al., 2024; Brownlee et al., 2023), Bayesian Optimization (Liu et al., 2024), and prompt optimization (Fernando et al., 2024; Guo et al., 2024), where unguided variations are sampled from LLMs. In contrast, LLEGO generates *guided* variations by considering search dynamics at the *population level*, modulating fitness and diversity through hyperparameters $\alpha$ and $\tau$, while exploiting in-context learning with parent solutions.

## 5 EXPERIMENTS

**Benchmark datasets.** We empirically evaluate LLEGO's ability to find performant decision trees for 12 open-source tabular datasets from OpenML curated benchmarks (Vanschoren et al., 2014), including 7 classification and 5 regression datasets. These datasets were selected based on the number of features, samples and the presence of semantically meaningful feature names and descriptions. We provide further details on this selection of datasets and preprocessing in Appendix C.1.

**Baselines.** We compare LLEGO against a comprehensive set of competitive decision tree induction methods across major categories: greedy induction (**CART** (Breiman, 2017) and **C4.5** (Quinlan, 1993)), sparse optimal tree induction (**GOSDT** (Lin et al., 2020) and **DL8.5** (Aglin et al., 2020)), and a GP approach using conventional genetic operators (**GATree** (Lahovnik, 2024), which is an implementation of GP for decision tree induction in Python). More details on these baselines, their implementation, hyperparameters, and experimental details are given in Appendices C.2 and C.3. For GP-based methods (**LLEGO**, **GATree**), we initialize the population with **CART** models bootstrapped on 25% of the training data. We report results using $G = 25$, $N = 25$, and we use $\alpha = 0.1$, $\tau = 10$ and $\nu = 4$ as the hyperparameters for the variation operators of LLEGO.

**Evaluation.** For classification tasks, we use balanced accuracy (**ACC**), and for regression tasks, mean squared error (**MSE**), computed on a held-out test dataset $\mathcal{D}_{test}$. Each metric is averaged over 5 runs with different random seeds, due to different dataset splits, and we present these averages with their standard deviations. For LLEGO, we use gpt-3.5-turbo version 0301 as the underlying LLM. For a fair comparison, each method is allowed 10 minutes of wall clock time per seeded run, which includes time spent on hyperparameter tuning.

### 5.1 LLEGO-EVOLVED TREES ACHIEVE SUPERIOR GENERALIZATION PERFORMANCE

We first compare the performance of the complete LLEGO algorithm against baselines for decision tree induction. We report in Table 1 and Table 2 the test performance on classification and regression datasets, respectively, for maximum tree depths of 3 and 4. For regression, we report the results for **CART** and **GATree** since other baselines cannot optimize regression objectives. The results demonstrate that LLEGO outperforms baselines comprehensively. We observe that this performance advantage becomes more pronounced in the space of trees with depth 4, which is intuitive since

Table 2: **Performance on regression tasks.** MSE (↓) across 5 regression datasets, best results emboldened.

| Method | Abalone | Cars | Cholesterol | Wage | Wine |
|--------|---------|------|-------------|------|------|
| | | *depth = 3* | | | |
| CART | $0.591_{(0.027)}$ | $0.250_{(0.028)}$ | $1.500_{(0.244)}$ | $\mathbf{1.036}_{(0.146)}$ | $\mathbf{0.811}_{(0.009)}$ |
| GATREE | $0.595_{(0.044)}$ | $\mathbf{0.198}_{(0.039)}$ | $1.427_{(0.187)}$ | $1.150_{(0.149)}$ | $0.825_{(0.016)}$ |
| LLEGO | $\mathbf{0.584}_{(0.030)}$ | $0.200_{(0.037)}$ | $\mathbf{1.324}_{(0.139)}$ | $1.045_{(0.149)}$ | $0.814_{(0.010)}$ |
| | | *depth = 4* | | | |
| CART | $0.561_{(0.018)}$ | $0.269_{(0.041)}$ | $1.552_{(0.230)}$ | $1.185_{(0.193)}$ | $\mathbf{0.807}_{(0.004)}$ |
| GATREE | $0.586_{(0.036)}$ | $0.100_{(0.020)}$ | $1.343_{(0.158)}$ | $1.188_{(0.168)}$ | $0.847_{(0.017)}$ |
| LLEGO | $0.577_{(0.029)}$ | $\mathbf{0.099}_{(0.020)}$ | $\mathbf{1.322}_{(0.145)}$ | $\mathbf{1.067}_{(0.203)}$ | $0.828_{(0.026)}$ |

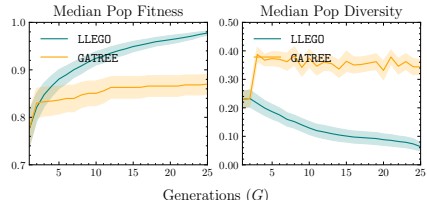

Figure 4: **Search efficiency.** Median fitness and diversity across 25 generations.

it represents a substantially larger search space compared to the set of trees with depth 3. In the more constrained space of trees with depth 3, sparse optimal induction methods such as **DL85** and **GOSDT** demonstrate increased competitiveness. This suggests that LLEGO's efficiency gains are particularly evident when navigating more complex and expansive search spaces. We also compare in Appendix D.1 the generalization gap across all methods. Notably, LLEGO consistently achieves the lowest generalization gap, with this advantage becoming especially pronounced at depth 4.

The results also underscore the impact of incorporating semantic priors. Indeed, LLEGO consistently outperforms the GP baseline **GATree**, which cannot take into account contextual information. Further analysis in Appendix D.5 demonstrates that LLEGO produces superior trees even when compared to a **GATree** configuration utilizing substantially larger search budgets. Notably, LLEGO achieves this superior performance while requiring fewer evaluations, highlighting its efficiency and effectiveness.

**Takeaway:** LLEGO optimizes decision trees that are superior against a diverse benchmark of methods, while being more applicable to a wider range of optimization objectives (e.g., regression).

**Search efficiency.** Having shown the superior generalization performance of decision trees evolved by LLEGO, we now compare search efficiency between LLEGO and the GP baseline **GATree**. We evaluate population dynamics via normalized population *fitness* and *diversity* between the two methods across all classification datasets, when optimizing trees with depth 3. Fitness values (i.e., balanced accuracy) were normalized to enable comparison across different seeds and datasets (refer to Appendix C.4 for details). Figure 4 (Left) shows the median population fitness, where LLEGO demonstrates superior search efficiency, finding *fitter* individuals more *efficiently*.

Figure 4 (Right) shows that the populations evolved by LLEGO exhibit decreasing diversity as the search progresses, whereas **GATree** maintains roughly the same level of diversity in its population. This is expected, as LLEGO uses its semantic priors to focus the search on semantically meaningful regions, which naturally reduces diversity. A similar effect has been observed when employing semantically aware GP in other domains (Krawiec & Pawlak, 2013). In comparison, **GATree**, which is semantically unaware, performs random structural perturbations that maintain a certain level of diversity in the population. In Appendix D.8, we investigate search efficiency on problems with depth 4 and show search dynamics on individual tasks, observing the same effects at play.

**Takeaway:** LLEGO leverages its semantic priors for more efficient search convergence, although this can sacrifice population diversity, requiring this trade-off to be carefully balanced by its operators.

## 5.2 UNDERSTANDING THE SOURCES OF GAIN

Having demonstrated enhanced search efficiency of LLEGO in the previous section, we now examine the contributions of its operators to this improvement. In what follows, we analyze how offspring characteristics are influenced by different values of the hyperparameters $\alpha$ and $\tau$, which give control over the desired solution fitness and population diversity.

**Results. (1) Crossover:** We examine the effect of $\alpha \in \{-0.25, -0.1, 0.1, 0.25\}$ on offspring generation, where $\alpha$ determines the target fitness $f^*$ that conditions the offspring generation. In Figure 5, we visualize the median population fitness and diversity as a function of $\alpha$. Offspring fitness improves as $\alpha$ increases from $-0.25$ to $0.1$, but regresses beyond this point as the target fitness $f^*$ leads to extrapolation in less reliable

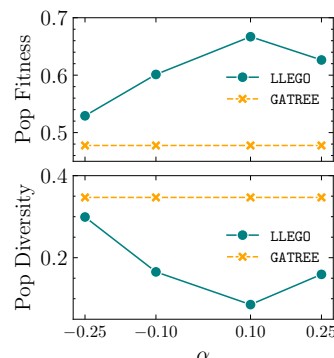

Figure 5: **XO dynamics.**

regions. Interestingly, the best offspring fitness emerges at $\alpha = 0.1$, suggesting LLEGO$_{\text{XO}}$'s ability to perform a reasonable degree of extrapolation. Corresponding, diversity decreases with increasing $\alpha$, reflecting sampling from progressively smaller search regions. Hence, compared to **GATree**, LLEGO produces higher quality offspring but with lower diversity, which is consistent with our findings in Section 5.1.

**(2) Mutation:** We investigate the role of LLEGO$_{\text{MUT}}$ in maintaining diversity by considering a range of $\tau \in \{5, 10, 25, 50\}$. In Figure 6, we observe that lower values of $\tau$ increases population diversity, as they prioritize offspring that have low likelihood given parents. As such, the offspring introduce greater diversity at the population level, which complements the dynamics of the crossover operator mentioned above, crucial in balancing exploitation and exploration during search. Results for individual datasets can be found in Appendix D.8.

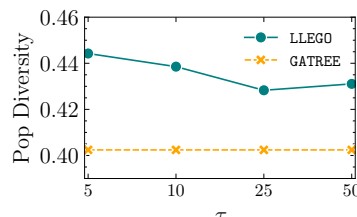

Figure 6: **MUT dynamics.**

### 5.3 ABLATION STUDY: ALL COMPONENTS CONTRIBUTE TO ENHANCED SEARCH EFFICIENCY

Having demonstrated the superior performance of LLEGO against existing baselines, we finally scrutinize the contribution of each algorithmic component to its optimization performance. Specifically, we aim to investigate the effects of **(1)** leveraging the LLM's semantic prior to evolve solutions, **(2)** the fitness-guided crossover and diversity-guided mutation, and **(3)** the higher arity of genetic operations. Now, we systematically ablate each component: LLEGO$_{\text{no\_semantics}}$ removes any semantic information from the prompts (see Appendix B.1 for detailed description), which is equivalent to removing $\mathcal{C}$ from $p(o|\mathcal{S}, \mathcal{C})$; LLEGO$_{\text{no\_xo}}$ removes the fitness-guided crossover, using only the mutation operator during search; LLEGO$_{\text{no\_mut}}$ removes

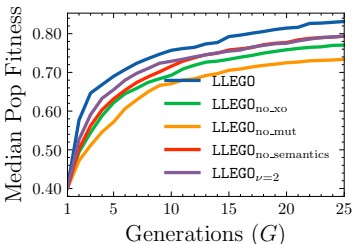

Figure 7: **Ablation study.** Comparing search efficiency of ablations.

diversity-guided mutation, using only crossover during search; and LLEGO$_{\nu=2}$ restricts the context to 2 parents, akin to traditional genetic operators. We evaluate search efficiency in Figure 7, observing that best performance is obtained when both operators are used in tandem, likely as they balance exploration of higher fitness regions (guided by $f^*$) and exploration of less visited regions (guided by $\tau$). The semantic information leveraged by the operator also improves performance, although we note that even without it, LLEGO$_{\text{no\_semantics}}$ performs very competitively, highlighting the strong few-shot learning capabilities of LLMs. Finally, using binary operators in LLEGO$_{\nu=2}$ is suboptimal, underlining the often overlooked importance of using a wider context in genetic operations. We provide more fine-grained ablation results in Appendix D.7.

**Takeaway:** Our ablation experiment demonstrates that all algorithmic components contribute to the enhanced optimization performance of LLEGO.

### 5.4 ADDITIONAL RESULTS.

In the interest of space, we relegated additional investigations to Appendix D. Specifically, we investigated *memorization concerns* by evaluating generalization performance on datasets with removed identifying information and context, as well as testing LLEGO on unseen proprietary datasets (detailed in Appendix D.2). In Appendix D.3, we investigated LLEGO's ability to mitigate negative bias by optimizing *fairness-regularized objectives*. We also provide additional analyses into LLEGO's performance and its individual components.

## 6 DISCUSSION

In summary, we introduced LLEGO, a novel GP method for decision tree induction that integrates semantic priors over the search space by using LLMs as variation operators. Our approach leverages the semantic understanding and domain knowledge of LLMs to evolve decision trees through crossover and mutation operators, while incorporating fitness and diversity guidance and flexible

operation arity. Empirical results across diverse datasets demonstrate LLEGO's superior optimization efficiency, yielding high-performing decision trees compared to existing baselines.

**Limitations and future works.** However, our work is not without its limitations. Performing inference through LLMs incurs a larger computational footprint than conventional GP algorithms. Our findings indicate that LLEGO trades off computational requirements for improved search efficiency and generalization performance, making it particularly appealing in performance-sensitive domains or problems where evaluation costs exceed search costs. Future works should prioritize reducing computational requirements while retaining performance, such as through inference acceleration (Leviathan et al., 2023) and memory-efficient model architectures (Han et al., 2015). Additionally, while LLEGO can operate effectively without semantic priors, its performance can be further improved when such knowledge is available. Future works could explore finetuning strategies and prompt augmentation strategies to incorporate semantic knowledge in specialized domains. We also recognize that using black-box LLMs could potentially lead to the propagation of negative biases into the solutions returned by LLEGO—to this end, we presented initial steps to mitigate bias via the design of objective functions. **Outlook.** In the long run, we believe this work illuminates the promise of employing LLM capabilities for enhancing efficiency and performance in complex combinatorial optimization problems beyond decision tree induction.

ETHICS AND REPRODUCIBILITY STATEMENTS

**Ethics.** In this work, we evaluate both public benchmarks and private datasets. The private datasets are *de-identified* and used following the guidance of the respective data providers. We follow recommendations to use the Azure OpenAI service when using GPT models, where via the agreement we ensure the medical data is not sent for human review or stored, hence respecting the guidelines given by the dataset providers.

**Reproducibility.** We provide all the details on the datasets, the implementation of baselines and the LLM in Appendix C. Furthermore, we detail the prompts used by the crossover and the mutation operators in Appendix B. We provide the code to reproduce our results at `https://github.com/nicolashuynh/LLEGO`, and `https://github.com/tennisonliu/LLEGO`.[1]

ACKNOWLEDGMENTS

We thank the anonymous ICLR reviewers, members of the van der Schaar lab, and Andrew Rashbass for many insightful comments and suggestions. TL would like to thank AstraZeneca for their sponsorship and support. NH thanks Illumina for their support. This work was supported by Microsoft's Accelerate Foundation Models Academic Research initiative.

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

# A ADDITIONAL DISCUSSIONS

## A.1 EXTENDED RELATED WORKS

**Tree induction algorithms.** Greedy algorithms sequentially grow trees by optimizing a given objective myopically. Popular methods in this class of algorithms are CART (Breiman et al., 1984), ID3 (Quinlan, 1986) and C4.5 (Quinlan, 1993). These algorithms differ in the predictive tasks in which they can be applied. These algorithms mainly differ in the criterion used to split the nodes at each local node, including Gini impurity (Breiman et al., 1984) or information gain (Quinlan, 1993). Owing to their greedy nature, they are computationally efficient in searching the combinatorial space. In contrast, a branch of work employs exact combinatorial optimization techniques to search for sparse, optimal trees, e.g. branch and bound (Lin et al., 2020) and dynamic programming (Aglin et al., 2020). Notable works include BinOCT (Verwer & Zhang, 2019), DL85 (Aglin et al., 2020), OSDT (Hu et al., 2019), and GOSDT (Lin et al., 2020). These approaches are fundamentally limited by the $NP$-hardness of the tree induction problem, and struggle to scale to larger size problems. Additionally, they have exclusively focused on the classification setting, and are limited in the types of feature (e.g. binary or continuous features) and objective functions that can be optimized. We compare LLEGO with representative tree induction methods in Table 3.

Table 3: **Comparison with the related works.** LLEGO provides a general framework for global optimization of decision trees, contrasting with prior works along several dimensions: computational complexity, support for different objective and regularization functions, task types, and incorporation of structural and semantic priors.

| Method | Algorithm | Worst-case complexity | Objective function | Arbitrary regularization | Task | | Priors | |
|---|---|---|---|---|---|---|---|---|
| | | | | | Classification | Regression | Structural | Semantic |
| CART (Breiman et al., 1984) | Greedy | $\mathcal{O}(2^h)$ | Gini impurity/MSE | ✗ | ✓ | ✓ | ✓ | ✗ |
| C4.5 (Quinlan, 1993) | Greedy | $\mathcal{O}(2^h)$ | Information gain | ✗ | ✓ | ✗ | ✓ | ✗ |
| DL8.5 (Aglin et al., 2020) | DP | $\mathcal{O}(d!)$ | Additive functions | ✗ | ✓ | ✗ | ✓ | ✗ |
| GOSDT (Lin et al., 2020) | DP | $\mathcal{O}(d!)$ | Monotonic functions | ✗ | ✓ | ✗ | ✓ | ✗ |
| LLEGO | GP | $\mathcal{O}(GN)$ | Any | ✓ | ✓ | ✓ | ✓ | ✓ |

**Genetic programming.** GP is an evolutionary optimization framework, particularly effective for a variety of combinatorial optimization problems, since it only requires the provision of a fitness function to evaluate and evolve a population of solutions to find optimal solutions (Koza, 1994a). As such, GP has been used in diverse tasks including tree induction (Tanigawa & Zhao, 2000; Kuo et al., 2007; Zhao, 2007; Koza, 1990), discovery symbolic mathematical expressions (Augusto & Barbosa, 2000; Qian et al., 2022), scheduling problems (Guillaume et al., 2007), neural architecture search (Broni-Bediako et al., 2020), and policy design (Hein et al., 2018). While the design of genetic operators differ significantly across domains, genetic operators share several limitations, being agnostic to the solution semantics, relying on stochastic perturbations without any search directionality, and narrow contexts. Several works in *semantic genetic programming* have considered the first two limitations and proposed variation operators (Krawiec & Pawlak, 2013; Moraglio et al., 2012) or rejection sampling mechanisms (Krawiec & Lichocki, 2009) to obtain semantic consistency between the offspring and their parents. However, these methods are domain-specific: for example, (Krawiec & Pawlak, 2013) considers convex combinations in the particular case of symbolic expressions. This limits their generalizability, and we note that no semantic operator has been designed for the tree induction setting which is the focus of our work.

**LLM and optimization.** Recent studies have explored LLMs for optimization tasks, exploiting their domain priors to enhance optimization efficiency (Song et al., 2024). Notable applications include prompt (Yang et al., 2024), reward-function (Ma et al., 2024), and code optimization (Liventsev et al., 2023). Particularly relevant is research employing LLMs as variation operators. (Lehman et al., 2023; Nasir et al., 2024; Brownlee et al., 2023) use LLMs as mutation operators for code

evolution, sampling mutation instructions from predefined sets. LLMs also have been utilized as variation operators for prompt optimization (Fernando et al., 2024), where task prompts contain explicit directives for generating variations. These approaches generate unguided variations, primarily utilizing LLMs' instruction-following capabilities. For example, in (Guo et al., 2024), crossover is performed using the prompt template: "Cross over the following prompts and generate a new prompt". Recent works have also considered the integration of LLMs with advanced evolutionary frameworks, namely quality-diversity algorithms (Pugh et al., 2016), to evolve both neural architectures and variation prompts (Nasir et al., 2024). In contrast, LLEGO generates guided variations, utilizing in-context learning of patterns in parent solutions to generate intelligent variations. Specifically, LLEGO steers offspring towards high-fitness regions by conditioning on desired fitness, while LLEGO controls diversity and exploration with the hyperparameter to define the offspring sampling distribution. Finally, recent work (Ye et al., 2024) has proposed using LLM for meta-heuristic optimization. It differs from LLEGO as it focuses on finding general meta-heuristics for a set of optimization tasks rather than tailoring the search with dataset-specific characteristics and relevant domain knowledge as LLEGO does.

### A.2 No Free Lunch

In accordance with the principle of No Free Lunch (Wolpert & Macready, 1997), we expect LLEGO to excel in domains with the following characteristics:

1. **Natural language representation:** Problems where solutions are expressible in natural language, enabling LLEGO to employ the LLM's semantic and contextual understanding for effective variations.
2. **Complex genotype-phenotype mapping:** Tasks with low locality, where LLEGO's semantic prior enhances variation efficacy.
3. **Contextual knowledge:** Domains benefiting from broader knowledge, where contextual knowledge (e.g. clinical guidelines for risk scoring) can be flexibly incorporated via prompt design ($\mathcal{C}$). This integration remains non-trivial for traditional evolutionary algorithms.
4. **Challenging operator design:** Areas where conventional semantic operators are difficult to craft (e.g. preserving semantics in program synthesis). LLEGO offers broadly applicable and flexibly customizable semantic variation operators.

These characteristics are prevalent in many applications, including decision trees, mathematical equations, and symbolic programs.

## B Complete Prompts

**Prompt design.** In this section, we describe the details of the prompts. To recap, each of the genetic operations is realized through natural language queries to the LLM. Each prompt is constructed of three essential elements:

1. **Task context.** This includes information about the input space $\mathcal{X}$, the output space $\mathcal{Y}$, and the characteristics of the dataset $\mathcal{D}$, e.g. number of samples, categorical features, continuous features. It also includes feature summary characteristics that are computed on the *training* set.
2. **Parent trees.** This contains the tree structure of each parent and possibly the fitness metric (in the case of crossover). These are translated to natural language and provided as few-shot examples to perform ICL in each genetic operation.
3. **Task-specific instructions.** For each genetic operator, we include task-specific instructions on offspring generations and guidelines on the format of the response.

The structured prompt for mutation is described in Figure 8. Descriptions enclosed in {}, such as {task_description} represent placeholder values that are populated dynamically at run-time. For a concrete example of this, the mutation prompt on credit dataset is shown in full in Listing 1. Similarly, the structured prompt for crossover is described in Figure 9 with a concrete example shown in Listing 2.

{task_description}. The dataset contains {n_samples} samples and {n_attributes} features, of which {n_numerical} are numerical and {n_categorical} are categorical. The target variable is {target_name}, it is {target_type}, {label_information}. The features and their ranges are: {feature_semantics}. You should generate a diverse decision tree that is more interpretable. Please generate decision trees in the desired JSON format, you can use any of the features, but are only allowed to use operators [<, >, <=, >=]. Return only the JSON in the format ## tree ##.

Figure 8: Prompt structure for **mutation operation**.

{task_description}. The dataset contains {n_samples} samples and {n_attributes} features, of which {n_numerical} are numerical and {n_categorical} are categorical. The target variable is {target_name}, it is {target_type}, {label_information}. The features and their ranges are: {feature_semantics}. Generate a different, interpretable decision tree which should have the improved fitness. Please generate decision trees in the desired JSON format, you can use any of the features, but are only allowed to use operators [<, >, <=, >=]. Return only the JSON in the format ## tree ##.

Figure 9: Prompt structure for **crossover operation**.

```
The task is to classify people described by a set of attributes as good
    or bad credit risks. The dataset contains 360 samples and 20
    features, of which 7 are numerical and 13 are categorical. The
    target variable is class, it is binary, the label distribution is
    [0: 29.17%, 1: 70.83%]. The features and their ranges are:
    [checking_status (int) [0, 3], duration (float) [5.00, 60.00],
    credit_history (int) [0, 4], purpose (int) [0, 9], credit_amount
    (float) [276.00, 15672.00], savings_status (int) [0, 4], employment
    (int) [0, 4], installment_commitment (float) [1.00, 4.00],
    personal_status (int) [0, 3], other_parties (int) [0, 2],
    residence_since (float) [1.00, 4.00], property_magnitude (int) [0,
    3], age (float) [19.00, 74.00], other_payment_plans (int) [0, 2],
    housing (int) [0, 2], existing_credits (float) [1.00, 4.00], job
    (int) [0, 3], num_dependents (float) [1.00, 2.00], own_telephone
    (int) [0, 1], foreign_worker (int) [0, 1]]. You should generate a
    diverse decision tree that is more interpretable. Please generate
    decision trees in the desired JSON format, you can use any of the
    features, but are only allowed to use operators [<, >, <=, >=].
    Return only the JSON in the format ## tree ##.

Expression: ## {'credit_history': {'<= 1.5000': {'property_magnitude':
    {'<= 0.5000': {'employment': {'<= 1.5000': {'value': 1}, '> 1.5000':
    {'value': 0}}}, '> 0.5000': {'value': 0}}}, '> 1.5000':
    {'savings_status': {'<= 3.5000': {'property_magnitude': {'<=
    0.5000': {'value': 0}, '> 0.5000': {'value': 1}}}, '> 3.5000':
    {'employment': {'<= 2.5000': {'value': 1}, '> 2.5000': {'value':
    1}}}}}}} ##
Expression: ## {'other_payment_plans': {'<= 1.5000':
    {'property_magnitude': {'<= 1.5000': {'own_telephone': {'<= 0.5000':
    {'value': 0}, '> 0.5000': {'value': 0}}}, '> 1.5000':
    {'num_dependents': {'<= 1.5000': {'value': 1}, '> 1.5000': {'value':
    0}}}}}, '> 1.5000': {'purpose': {'<= 6.5000': {'residence_since':
    {'<= 1.5000': {'value': 1}, '> 1.5000': {'value': 1}}}, '> 6.5000':
    {'housing': {'<= 0.5000': {'value': 1}, '> 0.5000': {'value':
    1}}}}}}} ##
Expression: ## {'credit_history': {'<= 3.5000': {'duration': {'<=
    34.5000': {'checking_status': {'<= 1.5000': {'value': 1}, '>
    1.5000': {'value': 1}}}, '> 34.5000': {'credit_amount': {'<=
    10552.5000': {'value': 1}, '> 10552.5000': {'value': 0}}}}}, '>
    3.5000': {'credit_amount': {'<= 9597.5000': {'employment': {'<=
```

```
    1.5000': {'value': 1}, '> 1.5000': {'value': 1}}}, '> 9597.5000':
    {'value': 0}}}}}} ##
Expression: ## {'property_magnitude': {'<= 0.5000': {'duration': {'<=
    33.0000': {'housing': {'<= 1.5000': {'value': 1}, '> 1.5000':
    {'value': 0}}}, '> 33.0000': {'employment': {'<= 0.5000': {'value':
    0}, '> 0.5000': {'value': 0}}}}}, '> 0.5000': {'employment': {'<=
    0.5000': {'credit_amount': {'<= 3359.5000': {'value': 0}, '>
    3359.5000': {'value': 1}}}, '> 0.5000': {'purpose': {'<= 5.5000':
    {'value': 1}, '> 5.5000': {'value': 1}}}}}}} ##
Expression: ##
```

Listing 1: **Example mutation prompt.** On `credit` dataset.

```
The task is to classify people described by a set of attributes as good
    or bad credit risks. The dataset contains 360 samples and 20
    features, of which 7 are numerical and 13 are categorical. The
    target variable is class, it is binary, the label distribution is
    [0: 29.17%, 1: 70.83%]. The features and their ranges are:
    [checking_status (int) [0, 3], duration (float) [5.00, 60.00],
    credit_history (int) [0, 4], purpose (int) [0, 9], credit_amount
    (float) [276.00, 15672.00], savings_status (int) [0, 4], employment
    (int) [0, 4], installment_commitment (float) [1.00, 4.00],
    personal_status (int) [0, 3], other_parties (int) [0, 2],
    residence_since (float) [1.00, 4.00], property_magnitude (int) [0,
    3], age (float) [19.00, 74.00], other_payment_plans (int) [0, 2],
    housing (int) [0, 2], existing_credits (float) [1.00, 4.00], job
    (int) [0, 3], num_dependents (float) [1.00, 2.00], own_telephone
    (int) [0, 1], foreign_worker (int) [0, 1]]. Generate a different,
    interpretable decision tree which should have the improved fitness.
    Please generate decision trees in the desired JSON format, you can
    use any of the features, but are only allowed to use operators [<,
    >, <=, >=]. Return only the JSON in the format ## tree ##.

fitness: 0.5882, Expression: ## {'purpose': {'<= 5.5000': {'housing':
    {'<= 0.5000': {'residence_since': {'<= 2.5000': {'value': 0}, '>
    2.5000': {'value': 1}}}, '> 0.5000': {'job': {'<= 1.5000': {'value':
    0}, '> 1.5000': {'value': 1}}}}}, '> 5.5000': {'duration': {'<=
    25.5000': {'credit_history': {'<= 3.5000': {'value': 1}, '> 3.5000':
    {'value': 1}}}, '> 25.5000': {'residence_since': {'<= 3.5000':
    {'value': 1}, '> 3.5000': {'value': 0}}}}}}} ##
fitness: 0.5930, Expression: ## {'savings_status': {'<= 2.5000':
    {'credit_amount': {'<= 9597.5000': {'credit_history': {'<= 0.5000':
    {'value': 0}, '> 0.5000': {'value': 1}}}, '> 9597.5000': {'value':
    0}}}, '> 2.5000': {'checking_status': {'<= 0.5000':
    {'property_magnitude': {'<= 0.5000': {'value': 0}, '> 0.5000':
    {'value': 1}}}, '> 0.5000': {'residence_since': {'<= 2.5000':
    {'value': 1}, '> 2.5000': {'value': 1}}}}}}} ##
fitness: 0.6162, Expression: ## {'property_magnitude': {'<= 0.5000':
    {'duration': {'<= 33.0000': {'housing': {'<= 1.5000': {'value': 1},
    '> 1.5000': {'value': 0}}}, '> 33.0000': {'employment': {'<=
    0.5000': {'value': 0}, '> 0.5000': {'value': 0}}}}}, '> 0.5000':
    {'employment': {'<= 0.5000': {'credit_amount': {'<= 3359.5000':
    {'value': 0}, '> 3359.5000': {'value': 1}}}, '> 0.5000': {'purpose':
    {'<= 5.5000': {'value': 1}, '> 5.5000': {'value': 1}}}}}}} ##
fitness: 0.6815, Expression: ## {'checking_status': {'<= 1.5000':
    {'property_magnitude': {'<= 1.5000': {'other_parties': {'<= 0.5000':
    {'value': 0}, '> 0.5000': {'value': 0}}}, '> 1.5000': {'duration':
    {'<= 20.5000': {'value': 1}, '> 20.5000': {'value': 1}}}}}, '>
    1.5000': {'credit_history': {'<= 2.5000': {'num_dependents': {'<=
    1.5000': {'value': 1}, '> 1.5000': {'value': 0}}}, '> 2.5000':
    {'other_payment_plans': {'<= 1.5000': {'value': 1}, '> 1.5000':
    {'value': 1}}}}}}} ##
fitness: 0.6908, Expression:
```

Listing 2: **Example crossover prompt.** On `credit` dataset.

**Tree representation.** We represent trees in natural language as a nested dictionary. This dictionary represents a decision tree where each key is a feature and the subsequent nested dictionaries correspond to decision rules and their outcomes. An example is illustrated in Figure 10 on the **iris** dataset. In this example, if 'petal width (cm)' is less than or equal to 0.80, the classification is 0; otherwise, further splits are made on 'petal width (cm)' at 1.75, leading to classifications of 1 or 2 depending on the condition.

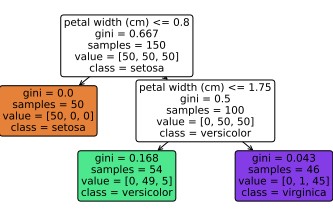

```
{
  "petal width (cm)": {
    "<= 0.80": {"value": 0},
    "> 0.80": {
      "petal width (cm)": {
        "<= 1.75": {"value": 1},
        "> 1.75": {"value": 2}
      }
    }
  }
}
```

Figure 10: **Example decision tree.** And its corresponding natural language representation as a nested dictionary.

### B.1 ABLATION PROMPTS

In our ablation study, we removed all semantic information from the prompts, with examples illustrated in Listing 3 and 4. Here, we remove the semantic description of the task, and replace its features names with $X_i$.

```
The task is to generate interpretable and high-performing decision trees
    given a set of attributes. The dataset contains 360 samples and 20
    features, of which 7 are numerical and 13 are categorical. The
    target variable is y, it is binary, the label distribution is [0:
    29.17%, 1: 70.83%]. The features and their ranges are: [X_0 (int)
    [0, 3], X_1 (float) [5.00, 60.00], X_2 (int) [0, 4], X_3 (int) [0,
    9], X_4 (float) [276.00, 15672.00], X_5 (int) [0, 4], X_6 (int) [0,
    4], X_7 (float) [1.00, 4.00], X_8 (int) [0, 3], X_9 (int) [0, 2],
    X_10 (float) [1.00, 4.00], X_11 (int) [0, 3], X_12 (float) [19.00,
    74.00], X_13 (int) [0, 2], X_14 (int) [0, 2], X_15 (float) [1.00,
    4.00], X_16 (int) [0, 3], X_17 (float) [1.00, 2.00], X_18 (int) [0,
    1], X_19 (int) [0, 1]]. You should generate a diverse decision tree
    that is more interpretable. Please generate decision trees in the
    desired JSON format, you can use any of the features, but are only
    allowed to use operators [<, >, <=, >=]. Return only the JSON in the
    format ## tree ##.

Expression: ## {'X_16': {'<= 1.5000': {'X_12': {'<= 38.5000': {'X_4':
    {'<= 2443.0000': {'value': 1}, '> 2443.0000': {'value': 0}}}, '>
    38.5000': {'X_1': {'<= 21.0000': {'value': 0}, '> 21.0000':
    {'value': 1}}}}}, '> 1.5000': {'X_0': {'<= 1.5000': {'X_3': {'<=
    5.5000': {'value': 0}, '> 5.5000': {'value': 1}}}, '> 1.5000':
    {'X_1': {'<= 19.0000': {'value': 1}, '> 19.0000': {'value': 1}}}}}}}
    ##
Expression: ## {'X_0': {'<= 0.5000': {'X_4': {'<= 976.5000': {'X_3':
    {'<= 3.5000': {'value': 0}, '> 3.5000': {'value': 0}}}, '>
    976.5000': {'X_5': {'<= 1.5000': {'value': 0}, '> 1.5000': {'value':
    1}}}}}, '> 0.5000': {'X_4': {'<= 13765.5000': {'X_12': {'<=
    22.5000': {'value': 0}, '> 22.5000': {'value': 1}}}, '> 13765.5000':
    {'value': 0}}}}} ##
Expression: ## {'X_5': {'<= 2.5000': {'X_4': {'<= 9597.5000': {'X_2':
    {'<= 0.5000': {'value': 0}, '> 0.5000': {'value': 1}}}, '>
    9597.5000': {'value': 0}}}, '> 2.5000': {'X_0': {'<= 0.5000':
    {'X_11': {'<= 0.5000': {'value': 0}, '> 0.5000': {'value': 1}}}, '>
    0.5000': {'X_10': {'<= 2.5000': {'value': 1}, '> 2.5000': {'value':
    1}}}}}}} ##
```

```
Expression: ## {'X_2': {'<= 0.5000': {'X_12': {'<= 23.5000': {'value':
    1}, '> 23.5000': {'value': 0}}}, '> 0.5000': {'X_5': {'<= 3.5000':
    {'X_12': {'<= 25.5000': {'value': 0}, '> 25.5000': {'value': 1}}},
    '> 3.5000': {'X_4': {'<= 1034.5000': {'value': 0}, '> 1034.5000':
    {'value': 1}}}}}}} ##
Expression: ##
```

Listing 3: **Example mutation prompt with semantics removed.** On `credit` dataset.

```
The task is to generate interpretable and high-performing decision trees
    given a set of attributes. The dataset contains 360 samples and 20
    features, of which 7 are numerical and 13 are categorical. The
    target variable is y, it is binary, the label distribution is [0:
    29.17%, 1: 70.83%]. The features and their ranges are: [X_0 (int)
    [0, 3], X_1 (float) [5.00, 60.00], X_2 (int) [0, 4], X_3 (int) [0,
    9], X_4 (float) [276.00, 15672.00], X_5 (int) [0, 4], X_6 (int) [0,
    4], X_7 (float) [1.00, 4.00], X_8 (int) [0, 3], X_9 (int) [0, 2],
    X_10 (float) [1.00, 4.00], X_11 (int) [0, 3], X_12 (float) [19.00,
    74.00], X_13 (int) [0, 2], X_14 (int) [0, 2], X_15 (float) [1.00,
    4.00], X_16 (int) [0, 3], X_17 (float) [1.00, 2.00], X_18 (int) [0,
    1], X_19 (int) [0, 1]]. Generate a different, interpretable decision
    tree which should have the improved fitness. Please generate
    decision trees in the desired JSON format, you can use any of the
    features, but are only allowed to use operators [<, >, <=, >=].
    Return only the JSON in the format ## tree ##.

fitness: 0.5882, Expression: ## {'X_3': {'<= 5.5000': {'X_14': {'<=
    0.5000': {'X_10': {'<= 2.5000': {'value': 0}, '> 2.5000': {'value':
    1}}}, '> 0.5000': {'X_16': {'<= 1.5000': {'value': 0}, '> 1.5000':
    {'value': 1}}}}}}, '> 5.5000': {'X_1': {'<= 25.5000': {'X_2': {'<=
    3.5000': {'value': 1}, '> 3.5000': {'value': 1}}}, '> 25.5000':
    {'X_10': {'<= 3.5000': {'value': 1}, '> 3.5000': {'value': 0}}}}}}}
    ##
fitness: 0.5930, Expression: ## {'X_5': {'<= 2.5000': {'X_4': {'<=
    9597.5000': {'X_2': {'<= 0.5000': {'value': 0}, '> 0.5000':
    {'value': 1}}}, '> 9597.5000': {'value': 0}}}, '> 2.5000': {'X_0':
    {'<= 0.5000': {'X_11': {'<= 0.5000': {'value': 0}, '> 0.5000':
    {'value': 1}}}, '> 0.5000': {'X_10': {'<= 2.5000': {'value': 1}, '>
    2.5000': {'value': 1}}}}}}} ##
fitness: 0.6162, Expression: ## {'X_11': {'<= 0.5000': {'X_1': {'<=
    33.0000': {'X_14': {'<= 1.5000': {'value': 1}, '> 1.5000': {'value':
    0}}}, '> 33.0000': {'X_6': {'<= 0.5000': {'value': 0}, '> 0.5000':
    {'value': 0}}}}}, '> 0.5000': {'X_6': {'<= 0.5000': {'X_4': {'<=
    3359.5000': {'value': 0}, '> 3359.5000': {'value': 1}}}, '> 0.5000':
    {'X_3': {'<= 5.5000': {'value': 1}, '> 5.5000': {'value': 1}}}}}}} ##
fitness: 0.6815, Expression: ## {'X_0': {'<= 1.5000': {'X_11': {'<=
    1.5000': {'X_9': {'<= 0.5000': {'value': 0}, '> 0.5000': {'value':
    0}}}, '> 1.5000': {'X_1': {'<= 20.5000': {'value': 1}, '> 20.5000':
    {'value': 1}}}}}, '> 1.5000': {'X_2': {'<= 2.5000': {'X_17': {'<=
    1.5000': {'value': 1}, '> 1.5000': {'value': 0}}}, '> 2.5000':
    {'X_13': {'<= 1.5000': {'value': 1}, '> 1.5000': {'value': 1}}}}}}}
    ##
fitness: 0.6908, Expression:
```

Listing 4: **Example crossover prompt with semantics removed.** On `credit` dataset.

## C   DETAILS OF EXPERIMENTAL PROCEDURES

In this section, we outline the benchmark datasets employed in our evaluations, as well as implementation details of our method and considered baselines.

## C.1 DATASET DETAILS

We employ a total of **12** datasets for our evaluation, of which 7 are classification tasks, and 5 are regression tasks. Additionally, we consider 2 propriety datasets in Appendix D.2, for which the LLM would not have seen during pretraining, and thus used to check for any memorization concerns.

**Open-source datasets.** The 12 open-source tabular datasets are sourced from OpenML (Vanschoren et al., 2014). The classification datasets were selected from the curated suite *OpenML-CC18* (Bischl et al., 2019) with the following criteria: $\leq 20$ features, $\leq 10000$ samples, binary labels and no missing data. This stems from the fact that optimal tree induction methods scale exponentially with the number of features and samples, and some baselines only support binary classification. Additionally, we excluded datasets lacking semantically meaningful feature names and descriptions, required by LLEGO. Regression datasets were selected from *OpenML-CTR23* (Fischer et al., 2023) with identical criteria. We detail dataset characteristics, including OpenML ID, number of attributes, number of samples and label distribution in Table 4. These datasets can be loaded by querying their OpenML IDs. The datasets describe:

- **credit** (Kelly et al.): This dataset classifies people as good or bad credit risks.
- **diabetes** (Smith et al., 1988): This dataset classifies patients based on WHO definition of diabetes.
- **compas** (Inc., 2016): Contains criminal history, jail and prison time, demographics, and is used to predict two year recidivism.
- **heart** (hea): Prediction of heart disease in patients.
- **liver** (Kelly et al.): This data set contains 416 liver patient records and 167 non liver patient records.The data set was collected from north east of Andhra Pradesh, India. The class label divides the patients into 2 groups (liver patient or not). This data set contains 441 male patient records and 142 female patient records.
- **heart** (Street et al., 1993): Features are computed from a digitized image of a fine needle aspirate (FNA) of a breast mass. They describe characteristics of the cell nuclei present in the image. The target feature records the prognosis (malignant or benign).
- **vehicle** (Pete & Shepherd): The dataset classifies a given silhouette as one of four types of vehicle, using a set of features extracted from the silhouette. The target label is re-relabelled, where the majority class as positive ('P') and all others as negative ('N').
- **cholesterol** (Janosi et al., 1988): The dataset predicts the cholesterol level among patients diagnosed with heart disease.
- **wine** (Cortez et al., 2009): The task is to predict quality of white and red wine.
- **wage** (Berndt, 1991): The task is to predict individual wages using the Current Population Survey (CPS), used to supplement census information between census years.
- **abalone** (Nash et al., 1995): Predicting the age of abalone from physical measurements. The age of abalone is determined by cutting the shell through the cone, staining it, and counting the number of rings through a microscope – a boring and time-consuming task.
- **cars** (Bohanec, 1997): Dataset of the suggested retail prices (column Price) and various characteristics of each car.

Table 4: **Open-source datasets.** Details of open-source datasets from OpenML (Vanschoren et al., 2014). **# Cat**: number of categorical attributes, **# Num**: number of numerical attributes, **Label dist**: label distribution.

| Dataset | ID | # Samples | # Attributes | # Num | # Cat | Label | Label distr |
|---|---|---|---|---|---|---|---|
| credit | 31 | 1000 | 20 | 7 | 13 | binary | 0: 29.17%, 1: 70.83% |
| diabetes | 37 | 768 | 8 | 8 | 0 | binary | 0: 66.30%, 1: 33.70% |
| compas | 42192 | 5278 | 13 | 5 | 8 | binary | 0: 52.50%, 1: 47.50% |
| heart | 53 | 270 | 13 | 5 | 8 | binary | 0: 52.58%, 1: 47.42% |
| liver | 1480 | 583 | 10 | 9 | 1 | binary | 0: 67.94%, 1: 32.06% |
| breast | 15 | 699 | 9 | 9 | 0 | binary | 0: 65.34%, 1: 34.66% |
| vehicle | 994 | 846 | 18 | 18 | 0 | binary | 0: 73.03%, 1: 26.97% |
| cholesterol | 204 | 303 | 13 | 6 | 7 | continuous | - |
| wine | 287 | 6497 | 11 | 11 | 0 | continuous | - |
| wage | 534 | 534 | 10 | 3 | 7 | continuous | - |
| abalone | 44956 | 4177 | 8 | 7 | 1 | continuous | - |
| cars | 44994 | 804 | 17 | 1 | 16 | continuous | - |

**Dataset preprocessing.** We preprocess the dataset using a train-validation-test split ratio of $[0.2, 0.4, 0.4]$. The low training split is used to accentuate the difference in performance as given sufficient training data, all methods perform comparably. For each run, we only vary the seed used for data splitting, such that for seed 0, we use `train_test_split(seed=0)`. For any algorithms that have inherent randomness (i.e. **CART** and **GATree**), we seed them with `seed=42`. As such, the randomness reported is induced only by different datasets.

We do not apply any additional preprocessing to continuous features. For categorical features, we follow the recommendations provided in §9.2.4 of (Hastie et al., 2009), where we rank each category of the predictor by calculating the proportion of observations that fall into the outcome class 1 (Hastie et al., 2009). This results in a ranking of the categories based on these proportions. No additional preprocessing is applied to categorical or continuous labels.

## C.2 IMPLEMENTATION DETAILS

**Baselines.** To assess the performance of `LLEGO`, we compare it against a comprehensive set of state-of-the-art algorithms, covering representative methods from main categories of tree induction. Specifically, **CART** and **C4.5** are *greedy* tree induction methods, **GOSDT** and **DL8.5** are *optimal* tree induction methods, and **GATree** is a *genetic programming* based approach:

- **CART** (Classification and Regression Trees) (Breiman et al., 1984): CART is a decision tree algorithm that splits data into subsets based on feature values, creating a binary tree for classification or regression tasks using measures like Gini impurity or mean squared error. We use the implementation provided in `sklearn.tree`, https://scikit-learn.org/stable/modules/generated/sklearn.tree.DecisionTreeClassifier.html.
- **C4.5** (Quinlan, 1993): C4.5 is an extension of the ID3 algorithm that generates decision trees by handling both categorical and continuous data, and uses information gain ratio to choose splits. We use the implementation provided in the PyPI package `c45-decision-tree`, https://pypi.org/project/c45-decision-tree/.
- **GOSDT** (Lin et al., 2020): GOSDT constructs decision trees by optimizing a trade-off between accuracy and complexity, ensuring sparsity and interpretability through global optimization techniques. We use the implementation provided by the original authors https://github.com/ubc-systopia/gosdt-guesses.
- **DL8.5** (Aglin et al., 2020): DL8.5 is a decision tree learning algorithm that focuses on constructing optimal decision trees given specific constraints, using dynamic programming to find the best tree structure. We use the implemented provided in the PyPI package `dl8.5`, https://github.com/ubc-systopia/gosdt-guesses.
- **GATree** (Lahovnik, 2024): GATree is a Python library designed for implementing evolutionary decision trees using a genetic algorithm approach. We use the official implementation https://gatree.readthedocs.io/en/latest/ and keep the defaults settings of the implementation (i.e. tournament selection, subtree crossover and subtree mutation).

**Hyperparameter search ranges.** Next, we detail the hyperparameters of each method, and their respective search ranges. Across experiments, we keep `max_depth` fixed to enable fair comparison, the details of tunable hyperparameters are detailed in Table 5.

**Hyperparameter tuning.** We use Optuna (Akiba et al., 2019) and the default Tree-Parzen Estimator for hyperparameter tuning (HPT) (Watanabe, 2023). For all baselines, we permit wall-clock time to a maximum of 10 minutes. This allows 50 iterations of HPT for **CART** and **C4.5**, and 10 iterations for the computationally more intensive **DL8.5**, **GOSDT**, and **GATree**. In each iteration of HPT, we evaluate the objective on the validation set, selecting the best configuration to evaluate on the test set.

**Computer resources.** We run all experiments on an `AMD EPYC 7V13 64-Core Processor`.

## C.3 `LLEGO` IMPLEMENTATION DETAILS

For our instantiation of `LLEGO` in Section 5, we use $N = 25$ and $G = 25$. We seed the algorithm with a population of trees generated by **CART**, where each tree is fitted on $25\%$ of the $\mathcal{D}_{\text{train}}$. We use the same population to initialize **GATree**. In each iteration, we generate 25 crossover offspring and 25 mutation offspring, using a rejection mechanism where invalid solutions are discarded (in Section 5, $\sim 86\%$ of crossover and $\sim 88\%$ of mutation offspring are syntactically valid). We use

Table 5: **Hyperparameter search ranges.** Hyperparameter search ranges for all baselines.

| | | |
|---|---|---|
| CART | min_samples_split | [int, 2, 16] |
| | min_samples_split | [int, 1, 16] |
| | max_depth | fixed |
| | splitter | best |
| | criterion | ['squared_error' (reg), 'gini' (clas)] |
| C4.5 | min_samples_split | [int, 2, 16] |
| | min_samples_split | [int, 1, 16] |
| | max_depth | fixed |
| DL8.5 | min_sup | [int, 1, 10] |
| | max_depth | fixed |
| GOSDT | regularization | [float, 0.001, 1] |
| | max_depth | fixed |
| GATree | population_size | [int, 10, 50] |
| | mutation_prob | [float, 0.1, 0.5] |
| | crossover_prob | [float, 0.1, 0.95] |
| | max_iterations | 100 |
| | tournament size | 2 |
| | max_depth | fixed |

elitism selection to preserve the top 25 trees after merging the offspring of the crossover and the mutation. To compute the desired fitness, we use $\alpha = 0.1$, based on observations in Section 5.2 as the value that balanced diversity and fitness. We use $\tau = 10$ for diversity guidance. For each genetic operation, we use $\lambda = 4$ parent trees. For our experiments, we use `gpt-35-turbo`, version `0301` with default hyperparameters `temperature` $= 0.7$ and `top_p` $= 0.95$.

**Function and terminal set.** For both **LLEGO** and **GATree**, the function set is $\{<, >, \leq, \geq\}$ and the terminal set includes numerical constants based on target feature values.

In Section 5.2, we perform 3 steps of crossover starting from the initial population, for both `LLEGO` and `GATree` to obtain Figure 5. We similarly perform 3 steps of mutation starting from the initial population to obtain Figure 6.

## C.4 Evaluation Metrics

**MSE.** For regression dataset, we report MSE (`sklearn.metrics.mean_squared_error`):

$$\text{MSE}(\mathcal{D}, f) = \frac{1}{N} \sum_{n=1}^{N} ||f(x_n) - y_n||^2$$

**Balanced accuracy.** For classification datasets, we report balanced accuracy, which is equivalent to accuracy with class-balanced sample weights (`sklearn.metrics.balanced_accuracy_score`). This has the effect of giving equal importance to both the positive and negative classes, thereby mitigating the impact of class imbalance and providing a more reliable assessment of the classifier's performance across all classes:

$$\text{BAcc} = \frac{1}{2} \left( \frac{TP}{TP + FN} + \frac{TN}{TN + FP} \right)$$

**Difference in equal opportunity.** When evaluating fairness, we consider *difference in equal opportunity* (DEO). This score measures the difference in recall between unprivileged and privileged groups, where a value of DEO $= 0$ indicates equality of opportunity.

$$\text{DEO} = |p(\hat{y} = 1 \,|\, \text{group} = 1, y = 1) - p(\hat{y} = 1 \,|\, \text{group} = 0, y = 1)|$$

We utilize the implementation `aif360.sklearn.metrics.equal_opportunity_difference` provided in https://aif360.readthedocs.io/ (Bellamy et al., 2019; Roemer & Trannoy, 2015).

**Population Fitness.** In order to assess the fitness of the populations evolved by the GP-based algorithms, we compute for a given population $\mathcal{P}$:

$$\text{Fitness} = \text{Median}(\{f'(t) \mid t \in \mathcal{P}\})$$

where $f'(t)$ denotes the normalized accuracy, calculated as $\frac{f(t) - \min_{t \in \mathcal{P}} f(t)}{\max_{t \in \mathcal{P}'} f(t) - \min_{t \in \mathcal{P}'} f(t)}$, where $f(t)$ here denotes the accuracy. $\mathcal{P}'$ is the union of all individuals produced by all methods for a particular seeded run on a particular dataset. In other words, the best accuracy obtained by *any* method on a particular seed for a particular dataset will have $f'(t) = 1$ and the worst will have $f'(t) = 0$. This normalization allows accuracy results from different datasets, seeds, and methods to be compared.

**Population diversity.** In order to assess the diversity of the populations evolved by the GP-based algorithms, we compute for a given population $\mathcal{P}$:

$$\text{Diversity} = \text{Median}(\{||\varphi(t) - \varphi(t')||_1 \mid (t, t') \in \mathcal{P}^2 \})$$

where $\varphi(t)$ denotes the functional signature of $t$, i.e. the vector $(t(\boldsymbol{x}_1), ..., t(\boldsymbol{x}_n))$.

# D    ADDITIONAL RESULTS

In this section of the appendix, we provide additional empirical results. Specifically:

1. In Appendix D.1, we analyze the train-test generalization gap.
2. In Appendix D.2, we report generalization performance on tasks with all semantics removed. The objectives of this experiment are to *(1)* check for memorization and *(2)* evaluate the contribution of semantic priors to search efficiency. We also evaluate LLEGO on proprietary datasets.
3. In Appendix D.3, we investigate the potential for bias and the flexibility of LLEGO in optimizing for fairness-regularized objectives.
4. In Appendix D.4, we demonstrate that the log-probabilities of candidate trees is directly correlated with the structural distances.
5. In Appendix D.5, we compare LLEGO against a version of GATree running with larger population sizes and more generations than LLEGO.
6. In Appendix D.6, we report the runtimes of LLEGO and the baselines.
7. In Appendix D.7, we perform additional ablation studies on key variables, including prompting strategies, choice of underlying LLM, genetic operation arity, and different parent sampling mechanisms, and population initialization strategies.
8. In Appendix D.8, we provide fine-grained results of the aggregate analysis presented in the main paper.

## D.1    GENERALIZATION GAP

In Table 6, we report the generalization gap (defined as $\frac{\text{BAcc}_{\text{train}} - \text{BAcc}_{\text{test}}}{\text{BAcc}_{\text{train}}}$) averaged across the classification datasets for each method. The results show that LLEGO consistently achieve a lower generalization gap compared to the baselines. In particular, the difference between LLEGO and the baselines is more noticeable at depth 4, where the baselines are more susceptible to overfitting, such as the optimal induction method DL85. This aligns with empirical observations in recent works (Zharmagambetov et al., 2021; Marton et al., 2023; Sullivan et al., 2024).

Table 6: **Relative generalization gap.** Averaged over the classification datasets.

| Method | Depth=3 | Depth=4 |
|--------|---------|---------|
| C45    | 0.073   | 0.086   |
| CART   | 0.078   | 0.101   |
| DL85   | 0.131   | 0.161   |
| GATREE | 0.086   | 0.092   |
| GOSDT  | 0.044   | 0.052   |
| LLEGO  | **0.043** | **0.043** |

## D.2 INVESTIGATION INTO MEMORIZATION

As with any LLM application, there is a concern about LLM memorization. Although it is highly unlikely that the LLM has encountered the optimal trees for the considered datasets—especially given that high-performing solutions can vary significantly across different training splits, seeds, and preprocessing steps—we empirically investigate this concern. This is done by removing any dataset-specific metadata or semantic information that could identify the underlying data. For prompts with semantics removed, please refer to Appendix B.1. We refer to this setting as $\texttt{LLEGO}_{\text{no\_semantics}}$ and compare its performance against $\texttt{LLEGO}$ with semantics included in Table 7. We observe that $\texttt{LLEGO}_{\text{no\_semantics}}$ achieves similar performance, even outperforming $\texttt{LLEGO}$ on two of the tasks.

Table 7: **Performance on classification tasks.** Comparing $\texttt{LLEGO}$ with $\texttt{LLEGO}_{\text{no\_semantics}}$ (i.e. all semantic information removed). Best results are emboldened.

| Method | Compas | Credit | Diabetes | Heart | Liver |
|---|---|---|---|---|---|
| *depth = 3* | | | | | |
| $\texttt{LLEGO}_{\text{no\_semantics}}$ | $\mathbf{0.654}_{(\mathbf{0.010})}$ | $\mathbf{0.683}_{(\mathbf{0.012})}$ | $0.700_{(0.033)}$ | $0.726_{(0.030)}$ | $0.643_{(0.033)}$ |
| $\texttt{LLEGO}$ | $0.652_{(0.004)}$ | $0.679_{(0.007)}$ | $\mathbf{0.713}_{(\mathbf{0.015})}$ | $\mathbf{0.736}_{(\mathbf{0.024})}$ | $\mathbf{0.672}_{(\mathbf{0.019})}$ |
| *depth = 4* | | | | | |
| $\texttt{LLEGO}_{\text{no\_semantics}}$ | $0.659_{(0.011)}$ | $0.667_{(0.024)}$ | $0.701_{(0.013)}$ | $0.716_{(0.038)}$ | $0.651_{(0.025)}$ |
| $\texttt{LLEGO}$ | $\mathbf{0.663}_{(\mathbf{0.005})}$ | $\mathbf{0.684}_{(\mathbf{0.011})}$ | $\mathbf{0.721}_{(\mathbf{0.017})}$ | $\mathbf{0.751}_{(\mathbf{0.042})}$ | $\mathbf{0.676}_{(\mathbf{0.021})}$ |

To further verify that $\texttt{LLEGO}$'s superior performance does not rely on memorization, we evaluate it on two proprietary datasets (requiring authorized access, and hence extremely unlikely to be in the LLM training corpus): MAGGIC (heart failure, (Wong et al., 2014)) and CUTRACT (prostate cancer, (CUTRACT, 2019)). We report the results against CART and GATree for depth $= 4$ in Table 8, showing that $\texttt{LLEGO}$ achieves superior performance on these private datasets, further demonstrating that it relies on generalized semantic priors rather than dataset-specific memorization.

Table 8: **Performance on proprietary datasets.** Comparing $\texttt{LLEGO}$ with CART and GATree, with depth $= 4$. Best results are emboldened.

| Method | MAGGIC | CUTRACT |
|---|---|---|
| CART | $0.610_{(0.014)}$ | $0.694_{(0.038)}$ |
| GATree | $0.619_{(0.015)}$ | $0.706_{(0.024)}$ |
| LLEGO | $\mathbf{0.623}_{(\mathbf{0.007})}$ | $\mathbf{0.710}_{(\mathbf{0.009})}$ |

## D.3 ADDRESSING BIAS VIA REGULARIZATION

As illustrated in the previous experiments, the genetic operators in $\texttt{LLEGO}$ benefit from the properties of LLMs (i.e. semantic priors and wide context). It is then natural to wonder if, conversely, negative artifacts of LLMs may propagate to the decision trees found by $\texttt{LLEGO}$.

**Setup.** In this experiment, we focus in particular on *bias*. More precisely, we assess group fairness (Verma & Rubin, 2018) by computing the Difference in Equality of Opportunity (DEO) metric, defined as the difference in recall between unprivileged and privileged groups (cf. Appendix C.4 for an exact definition). We show an illustrative example on the dataset COMPAS, which is known to be biased on the sensitive attribute *race African American* (Angwin et al., 2016). Our objective is to mitigate bias with a DEO-based regularization, by defining $\texttt{LLEGO}$'s new fitness function, i.e. $f'(t) = f(t) + \beta \text{DEO}(t)$ for any $t \in \mathcal{T}$.

Table 9: **Fairness aware objective.**

| Method | FA? | Compas(race) | |
|---|---|---|---|
| | | ACC ($\uparrow$) | DEO ($\downarrow$) |
| CART | ✗ | $0.651_{(0.012)}$ | $0.255_{(0.016)}$ |
| C4.5 | ✗ | $0.650_{(0.008)}$ | $0.258_{(0.014)}$ |
| DL85 | ✗ | $\mathbf{0.666}_{(\mathbf{0.006})}$ | $0.264_{(0.008)}$ |
| GOSDT | ✗ | $0.641_{(0.003)}$ | $0.187_{(0.019)}$ |
| LLEGO | ✗ | $0.652_{(0.004)}$ | $0.308_{(0.070)}$ |
| LLEGO | ✓ | $0.651_{(0.002)}$ | $\mathbf{0.161}_{(\mathbf{0.071})}$ |

**Results.** As can be seen in Table 9, $\texttt{LLEGO}$ does not natively return fair decision trees when the fitness functions are based purely on accuracy. However, the DEO regularization permits $\texttt{LLEGO}$ to find decision trees with less bias compared to the other baselines. This highlights the flexibility

of `LLEGO`, which can handle composite search objectives unlike the other baselines. `LLEGO` also returns a population of individuals, which makes it possible to trade-off predictive performance with fairness metrics. We show this in Figure 11, where one can choose individuals returned by `LLEGO` with acceptable tradeoffs.

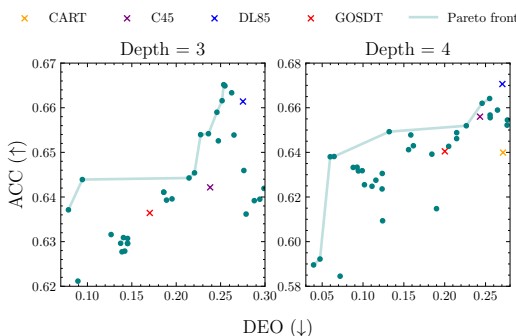

Figure 11: **Accuracy-fairness tradeoff.** On `compas` dataset.

### D.4 CORRELATION BETWEEN LOG-PROBABILITIES AND TREE EDIT DISTANCE

We show in this experiment that the log-probabilities of the offspring trees (utilized in `LLEGO`'s mutation operator) are negatively correlated with the structural distances between parent and offspring solutions.

**Experimental setting.** We generate 1000 offspring trees using the `LLEGO`'s mutation operator with a single parent tree. For each offspring individual, we assess its structural distance to the parent tree by computing the Tree Edit Distance (TED) (Bille, 2005) between this individual and the parent.

**Observations.** As shown in Figure 12, we observe a strong negative correlation (correlation coefficient = $-0.85$) between log-probabilities of the offspring and TED values. This relationship indicates that offspring with lower log-probabilities tend to exhibit greater structural differences from their parent, as measured by the TED. This demonstrates that `LLEGO`'s log-probability-based selection mechanism inherently promotes diversity in the population by favoring mutations which introduce varied structural changes.

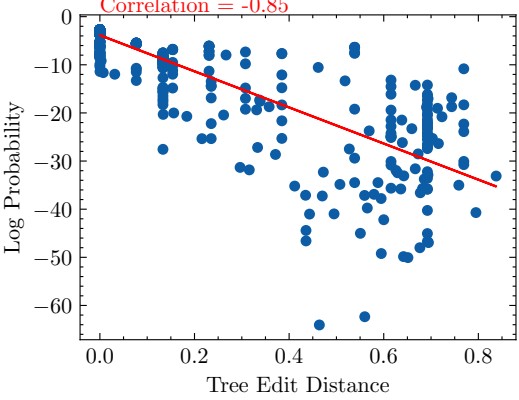

Figure 12: **Correlation between log-probabilities and stuctural distances.** There is a strong negative correlation between log-probabilities of the offspring and their TED values with respect to the parent tree.

## D.5 Additional Comparison With GATree

We extend our comparisons against GATree by increasing the population size to $N = 100$ and the number of generations to $G = 200$, while keeping LLEGO's default hyperparameters. We report the results for the classification and regression tasks in Table 10 and Table 11. Despite GATree's larger number of evaluations, LLEGO evolved superior trees. This underscores the importance of LLEGO's integration of semantic priors, search guidance, and broader context to enhance search efficiency. This superior search efficiency is especially important in settings where evaluation costs significantly exceed search costs (e.g. complex simulations, hardware optimizations, robotics control).

Table 10: **Comparison against `GATree`.** Balanced accuracy ($\uparrow$) on classification tasks (depth $d = 4$).

| Method | Breast | Compas | Credit | Diabetes | Heart | Liver | Vehicle |
|---|---|---|---|---|---|---|---|
| GATREE ($N = 100, G = 200$) | $0.948_{(0.011)}$ | $0.658_{(0.003)}$ | $0.667_{(0.009)}$ | $0.684_{(0.013)}$ | $0.738_{(0.028)}$ | $0.635_{(0.019)}$ | $\mathbf{0.939}_{(\mathbf{0.017})}$ |
| LLEGO ($N = 25, G = 25$) | $\mathbf{0.951}_{(\mathbf{0.007})}$ | $\mathbf{0.663}_{(\mathbf{0.005})}$ | $\mathbf{0.684}_{(\mathbf{0.011})}$ | $\mathbf{0.721}_{(\mathbf{0.017})}$ | $\mathbf{0.751}_{(\mathbf{0.042})}$ | $\mathbf{0.676}_{(\mathbf{0.021})}$ | $0.929_{(0.015)}$ |

Table 11: **Comparison against `GATree`.** MSE ($\downarrow$) on regression tasks (depth $d = 4$).

| Method | Abalone | Cars | Cholesterol | Wage | Wine |
|---|---|---|---|---|---|
| GATREE ($N = 100, G = 200$) | $\mathbf{0.566}_{(\mathbf{0.022})}$ | $0.099_{(0.012)}$ | $1.395_{(0.202)}$ | $1.143_{(0.147)}$ | $0.829_{(0.027)}$ |
| LLEGO ($N = 25, G = 25$) | $0.577_{(0.029)}$ | $0.099_{(0.025)}$ | $\mathbf{1.322}_{(\mathbf{0.145})}$ | $\mathbf{1.067}_{(\mathbf{0.203})}$ | $\mathbf{0.828}_{(\mathbf{0.026})}$ |

## D.6 Run-time Comparisons

We provide the total runtimes for the different methods in Table 12, averaged across the 7 classification datasets used in Section 5.1. We also report in Table 13 the detailed timings for LLEGO and GATREE with varying population sizes ($P \in \{25, 100\}$) and generations ($G \in \{25, 100, 200\}$), and also report the number of functional evaluations. These results, along with the ones presented in Section 5.1, highlight that LLEGO evolves superior trees compared to GATREE while necessitating less functional evaluations and wall-clock time. Nevertheless, we acknowledge that there is room for improvement for the runtime of LLEGO. Potential solutions include (1) reducing runtime through inference acceleration techniques such as speculative decoding and vLLM serving (Leviathan et al., 2023) and (2) reducing memory requirements through specialized fine-tuned models or quantization (Han et al., 2015).

Table 12: **Runtime comparisons (all methods).** Total runtime (in seconds), averaged across 7 classification datasets.

| | CART | C4.5 | DL85 | GOSDT | GATREE | LLEGO |
|---|---|---|---|---|---|---|
| Total run time (depth $d = 3$) | 0.0022 | 0.08 | 22.10 | 261.14 | 15.50 | 407.66 |
| Total run time (depth $d = 4$) | 0.0023 | 0.13 | 172.70 | 234.44 | 15.77 | 430.32 |

Table 13: **Runtime comparisons (GP methods).** Per-generation, total runtime (in seconds), and total number of fitness evaluations (depth $d = 4$, averaged across 7 classification datasets).

| | Per-generation runtime | Total run-time | # Functional evaluations |
|---|---|---|---|
| GATREE ($N = 25, G = 25$) | 0.63 | 15.77 | 620 |
| GATREE ($N = 100, G = 100$) | 2.65 | 264.95 | 9600 |
| GATREE ($N = 100, G = 200$) | 3.86 | 772.97 | 19200 |
| LLEGO ($N = 25, G = 25$) | 17.22 | 430.32 | 1250 |

## D.7 Additional Ablation Results

This subsection performs additional investigations into several key variables affecting LLEGO performance. We investigate the impact of diverse prompting strategies (Appendix D.7.1), the selection of different LLMs as genetic operators (Appendix D.7.2), and the impact of arity of genetic operations (Appendix D.7.3). Additionally, we analyze how various parent sampling mechanisms for crossover and mutation influence outcomes (Appendices D.7.4 and D.7.5), alongside an evaluation of different population initialization strategies (Appendix D.7.6).

### D.7.1 Prompting Strategies

**Experimental setting.** We compare LLEGO to LLEGO$_{\text{naive}}$, a variant which removes the crossover operator and changes the mutation prompt to an "improve the solution"-type of prompt.

**Results.** We report the results in Table 14, where we see that LLEGO consistently outperforms LLEGO$_{\text{naive}}$. This demonstrates the importance of explicit fitness-guidance via the hyperparameter $\alpha$ in order to steer the search towards high-fitness regions.

Table 14: **Performance of naive prompting.** Test balanced accuracy ($\uparrow$) on classification tasks (depth $d = 4$, 3 seeds), reporting mean$_{\text{(std)}}$.

| Method | Breast | Compas | Credit | Diabetes | Heart | Liver | Vehicle |
|---|---|---|---|---|---|---|---|
| LLEGO$_{\text{naive}}$ | $0.942_{(0.006)}$ | $0.660_{(0.011)}$ | $0.670_{(0.003)}$ | $0.708_{(0.019)}$ | $0.714_{(0.051)}$ | $0.629_{(0.033)}$ | $\mathbf{0.943_{(0.015)}}$ |
| LLEGO | $\mathbf{0.951_{(0.007)}}$ | $\mathbf{0.663_{(0.005)}}$ | $\mathbf{0.684_{(0.011)}}$ | $\mathbf{0.721_{(0.017)}}$ | $\mathbf{0.751_{(0.042)}}$ | $\mathbf{0.676_{(0.021)}}$ | $0.929_{(0.015)}$ |

### D.7.2 DIFFERENT LLMS

A key property of LLEGO's design is that it is LLM-agnostic. To demonstrate the advantage of this flexibility, we evaluate LLEGO's performance using gpt-4, comparing it to gpt-3.5. We report the results in Table 15 for all the classification datasets, for depth 4 problems, across 3 seeds. We see that the gpt-4 variant of LLEGO achieves superior performance than its gpt-3.5 counterpart. These results have two significant implications, as they indicate that (1) LLEGO's effectiveness is robust across LLM architectures, and importantly that (2) its performance can scale with advances in capabilities of the underlying LLMs.

Table 15: **Performance of different LLMs.** Test balanced accuracy ($\uparrow$) on classification tasks (depth $d = 4$, 3 seeds), reporting mean$_{\text{(std)}}$.

| Method | Breast | Compas | Credit | Diabetes | Heart | Liver | Vehicle |
|---|---|---|---|---|---|---|---|
| LLEGO (gpt-35) | $0.951_{(0.007)}$ | $0.663_{(0.005)}$ | $0.684_{(0.011)}$ | $0.721_{(0.017)}$ | $0.751_{(0.042)}$ | $\mathbf{0.676_{(0.021)}}$ | $0.929_{(0.015)}$ |
| LLEGO (gpt-4) | $\mathbf{0.957_{(0.005)}}$ | $\mathbf{0.671_{(0.011)}}$ | $0.684_{(0.008)}$ | $\mathbf{0.741_{(0.023)}}$ | $0.751_{(0.017)}$ | $0.640_{(0.025)}$ | $\mathbf{0.951_{(0.015)}}$ |

### D.7.3 ARITY OF GENETIC OPERATIONS

In Figure 5, we compared the crossover dynamics between LLEGO$_{XO}$ with $\nu = 4$ parents and roulette wheel selection, and GATree$_{XO}$ with $\nu = 2$ parents and uniform parent sampling. In Figure 13 (*Left*), we compare LLEGO$_{XO}$ with $\nu = 2$ parents and uniform parent sampling against GATree$_{XO}$ with $\nu = 2$ parent and uniform parent sampling. In Figure 13 (*Right*), we compare LLEGO$_{XO}$ with $\nu = 4$ parents and uniform parent sampling against GATree$_{XO}$ with $\nu = 2$ parent and uniform parent sampling.

We observe similar dynamics as in Figure 5, where varying $\alpha$ enables to control the population fitness and diversity. Additionally, $\nu = 4$ leads to significantly improved offspring fitness at the cost of a lower diversity, highlighting the nuanced impact of higher arity on search efficiency (corroborating the ablation results in Figure 7).

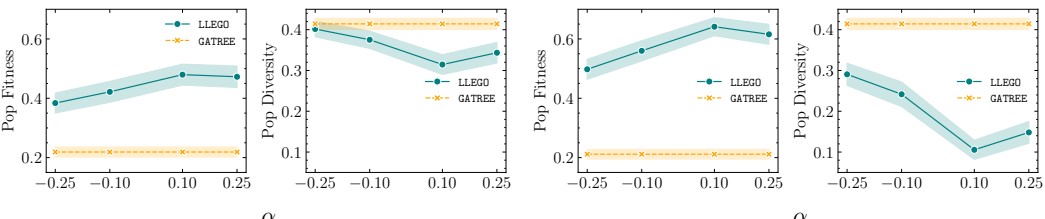

Figure 13: **XO dynamics.** Effect of fitness guidance ($\alpha$) on population and diversity using uniformly sampled parents. (**Left**) $\nu = 2$ parents, (**Right**) $\nu = 4$ parents

### D.7.4 PARENT SAMPLING: CROSSOVER

The objective of this experiment is to analyze the impact of an alternative selection mechanism on the balance between population fitness and diversity in the crossover operator.

**Experimental setting.** Specifically, we replace the roulette wheel selection (fitness-proportionate) mechanism with a tournament selection mechanism (Miller et al., 1995) with varying tournament

sizes $k \in \{1, 2, 3, 5\}$. We then compute the median offspring fitness and diversity as a function of $k$, following the experimental setup described in Section 5.2.

**Observations.** The results, shown in Figure 14, demonstrate a clear trade-off between fitness and diversity which is modulated by the tournament size. As shown in Figure 14a, larger tournament sizes consistently yield higher population fitness, while Figure 14b shows a corresponding decrease in population diversity. Indeed, larger values of $k$ intensify selection pressure by increasing the probability that highly fit individuals win multiple tournaments, thereby reducing population diversity. Conversely, smaller values of $k$ lead to an increased population diversity. For example, when $k = 1$, tournament selection corresponds to random sampling, which maximizes diversity at the cost of fitness performance. In comparison to tournament selection, the roulette wheel selection mechanism employed in LLEGO achieves a good middle-ground by striking a balance between fitness and diversity.

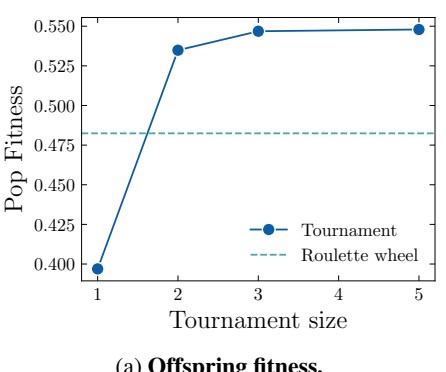

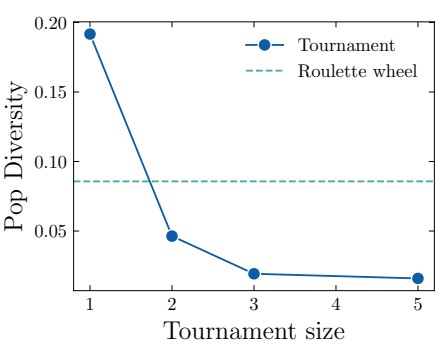

(a) **Offspring fitness.**      (b) **Population diversity.**

Figure 14: **Crossover dynamics with tournament selection.** (a) Population fitness increases monotonically with tournament size, demonstrating enhanced selective pressure. (b) Population diversity exhibits an inverse relationship with tournament size, with smaller tournaments preserving higher diversity at the cost of reduced fitness.

### D.7.5   PARENT SAMPLING: MUTATION

In this experiment, we investigate an alternative choice for the selection mechanism in LLEGO's mutation operator.

**Experimental setting.** We replace the random parent selection in the mutation operator with the quality-diversity algorithm CVT-MAP-Elites (Vassiliades et al., 2017), which requires defining a behavioral space. Given $n$ training samples, we define the behavioral space for classification tasks as $\mathcal{H} = [0, 1]^n$, encompassing the trees' functional signatures. The CVT-MAP-Elites algorithm then partitions $\mathcal{H}$ into $M$ niches using uniformly distributed centroids found with k-means clustering. We then select parents for the mutation operator by uniformly sampling $\nu$ niches and selecting the best individual in the sampled niches. Finally, we compute the offspring diversity, similarly as in Section 5.2.

**Observations.** We report the results in Figure 15, averaged across the classification datasets. We see that the total number of niches $M$ serves as a control parameter for offspring diversity, with an increasing relationship between diversity and the number of niches $M$. When $M = 1$, the process reduces to repeatedly sampling the population's best individual, resulting in minimal diversity for the generated offspring. Conversely, when solutions are spread into distinct niches, the sampling process becomes equivalent to uniform sampling without replacement from the population, yielding high diversity. Furthermore, we see in Figure 15 that the random selection of parents employed in LLEGO comparatively yields high diversity, justifying its use in the diversity-guided mutation operator.

### D.7.6   POPULATION INITIALIZATION STRATEGIES

In this experiment, we investigate the impact of a different population initialization on the search performance of LLEGO.

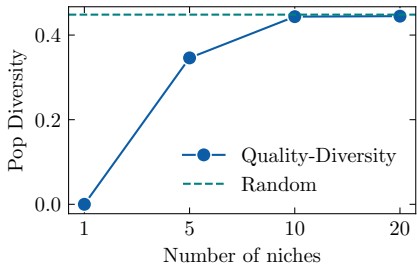

Figure 15: **Mutation dynamics with Quality-Diversity selection.** Offspring diversity increases with the number of niches employed in CVT-MAP-Elites for parent selection.

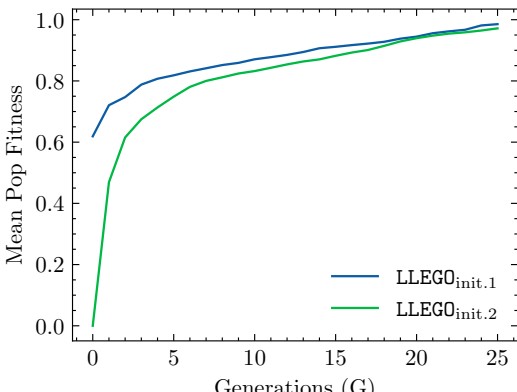

Figure 16: **Ablation on the population initialization.** LLEGO$_{\text{Init. 1}}$ initializes populations using CART models trained on 25% bootstrap samples, while LLEGO$_{\text{Init. 2}}$ uses minimal training subsets of size 2. Results are aggregated across all classification datasets for one seed.

**Experimental setting.** Specifically, we compare two variants of LLEGO. The baseline variant, LLEGO$_{\text{Init. 1}}$ corresponds to the instanciation of LLEGO described in Section 5, which initializes the population with CART models trained on bootstrap samples comprising 25% of the training data. In contrast, LLEGO$_{\text{Init. 2}}$ initializes trees using CART models trained on minimal random subsets of just two training samples, resulting in weaker initial decision trees.

**Observations.** Figure 16 illustrates the convergence of the mean population fitness across generations, aggregated and normalized over all classification datasets for one random seed. The results demonstrate that LLEGO$_{\text{Init. 2}}$ exhibits slower convergence compared to LLEGO$_{\text{Init. 1}}$, which shows the role of effective population initialization in improving search efficiency and faster discovery of high-quality solutions. Nevertheless, we remark that LLEGO$_{\text{Init. 2}}$ still achieves good performance in the later stages of the search (after $G = 20$ generations), showing the effectiveness of LLEGO's variation operators in steering the search towards promising regions, independent of the initialization scheme.

### D.8 FINE-GRAINED RESULTS

In this subsection, we present a detailed analysis of the results from the main paper. Our examination includes: (1) mutation dynamics observed in individual tasks; (2) convergence analysis across varying depths, including ablation studies; and (3) convergence trajectories for specific tasks.

**Mutation dynamics.** We provide the mutation dynamics for each individual dataset in Figure 17, showing that $\tau$ meaningfully controls the diversity in the population for 5 of the 7 classification datasets, where the diversity metrics are computed between parents and offspring (*Top*) and among the offspring (*Bottom*).

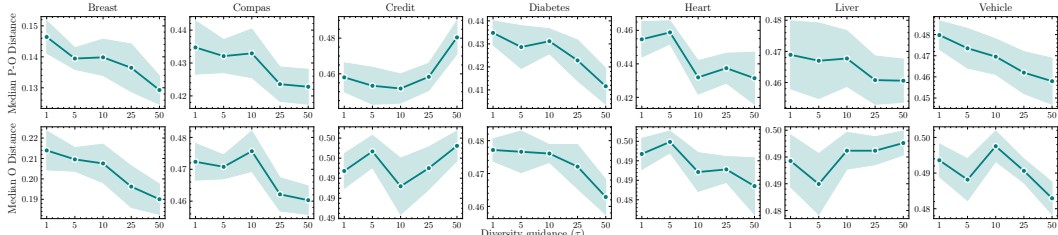

Figure 17: **MUT dynamics.** Effect of diversity guidance ($\tau$) on **(Top)** median parent-offspring distance and **(Bottom)** median offspring distance.

**Convergence analysis.** We provide separate convergence plots in this subsection, obtained when optimizing trees of depths 3 and 4, under the experimental setup described in Section 5.1. The results are reported in Figure 18a and Figure 18b. In these two settings, LLEGO leads to a more efficient search compared to GATree. This improved efficiency also comes with a reduced diversity, showing that LLEGO concentrates its populations in the later generations in high-fitness regions.

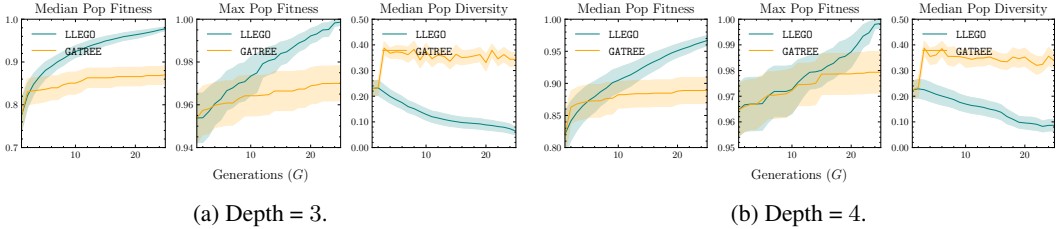

(a) Depth = 3.  (b) Depth = 4.

Figure 18: **Convergence dynamics.** Comparing LLEGO with GATREE.

**Ablation study.** We report the ablation study results for depth 3 and 4 in Figure 19 and Figure 20. These results align with the observations made in Section 5.3, highlighting the importance of using crossover and mutation in tandem, the importance of incorporating more than 2 parents for the operators and using semantic information. With a higher maximum depth, the space of possible trees becomes more complex, and accentuates the need for both exploration and exploitation, which explains why the mutation only (LLEGO$_{\text{no\_xo}}$) and crossover only (LLEGO$_{\text{no\_mut}}$) baselines perform worse than LLEGO.

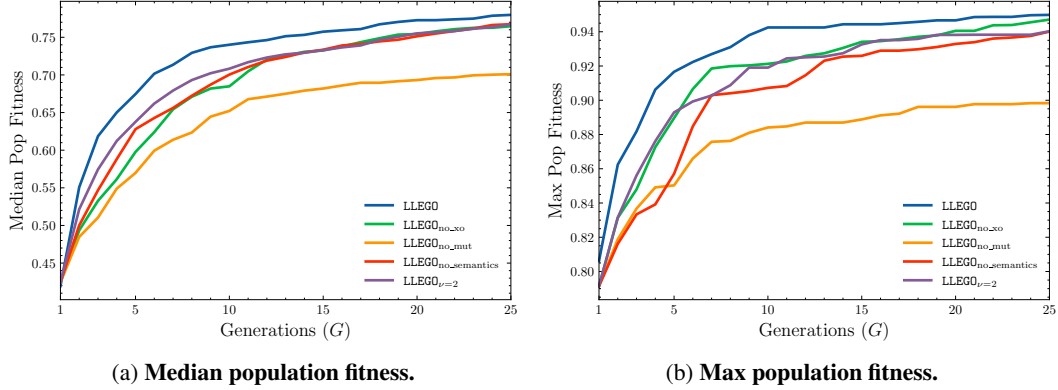

(a) **Median population fitness.**  (b) **Max population fitness.**

Figure 19: **Additional ablation results.** Depth = 3.

**Individual task results.** Convergence plots comparing LLEGO and **GATree** for individual tasks are given in Figure 21 and Figure 22. They show that LLEGO consistently leads to better search efficiency compared to GAtree.

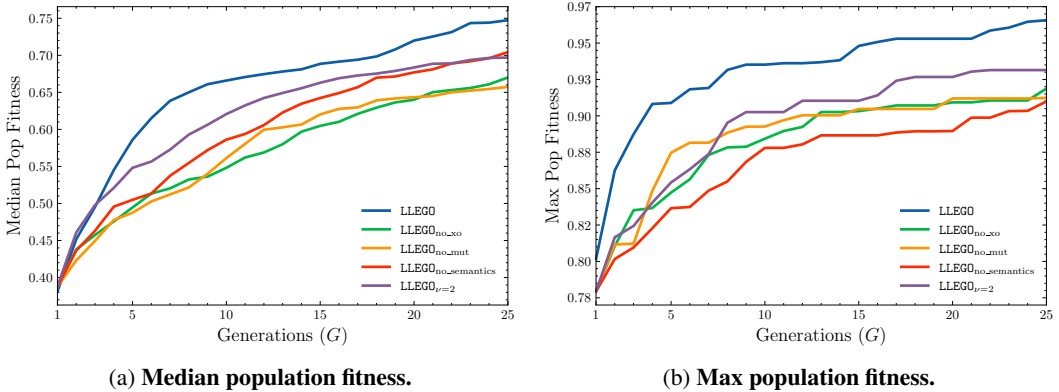

(a) **Median population fitness.**    (b) **Max population fitness.**

Figure 20: **Additional ablation results.** Depth = 4.

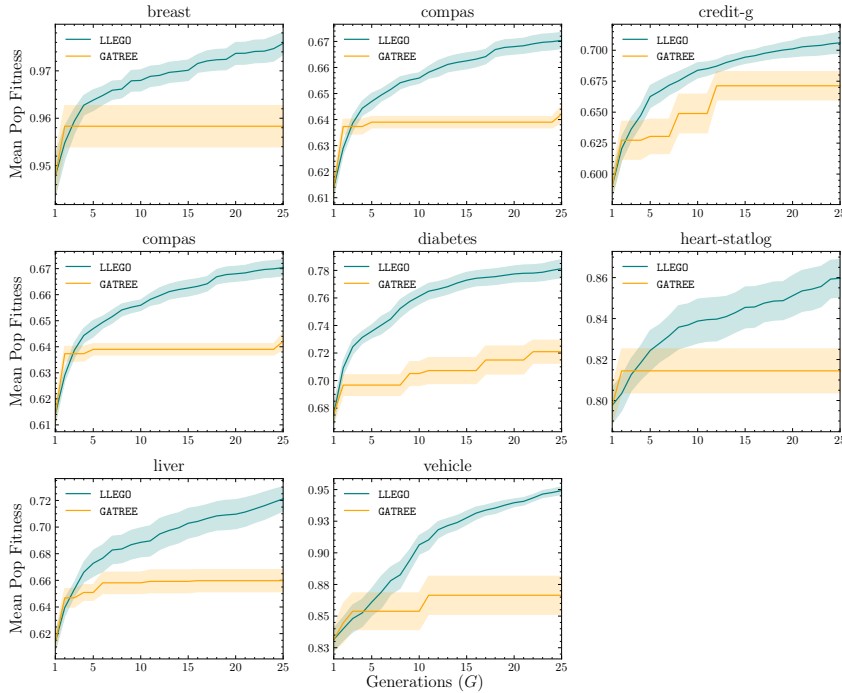

Figure 21: **Convergence plots.** Mean population fitness (↑) of `LLEGO` and `GATREE` on individual tasks across 25 generations (depth=3).

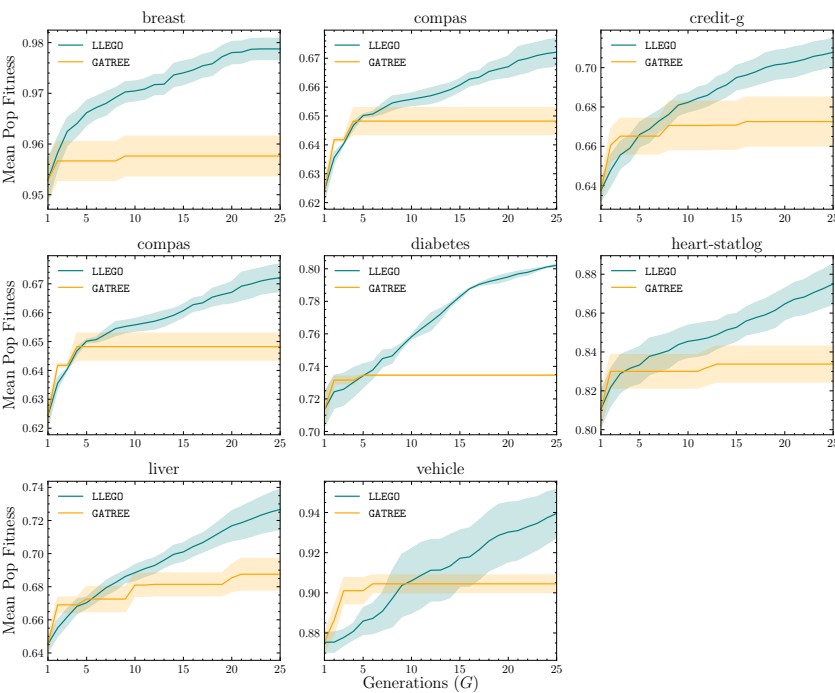

Figure 22: **Convergence plots.** Mean population fitness (↑) of LLEGO and GATREE on individual tasks across 25 generations (depth=4).

