# OpenReview forum: "Decision Tree Induction Through LLMs via Semantically-Aware Evolution"
_ICLR.cc/2025/Conference — ICLR 2025 Poster_

### Official Review · Reviewer_BF7z · 2024-10-21

**Soundness:** 3
**Presentation:** 3
**Contribution:** 3
**Rating:** 6
**Confidence:** 4

**Summary:**

The paper introduces a novel framework LLEGO that leverages LLMs to enhance genetic programming (GP) for decision tree induction. LLEGO incorporates semantic priors and domain knowledge into the optimization process through LLM-based fitness-guided crossover and diversity-guided mutation operators. The framework represents decision trees in natural language, enabling the use of broader contexts and higher arity operations. Empirical results on various benchmarks demonstrate LLEGO's superior performance over existing tree induction methods in terms of search efficiency and the quality of the evolved decision trees.

**Strengths:**

- Combining LLM and EA for decision tree induction is novel.
- The introduction of fitness-guided crossover and diversity-guided mutation is well-motivated.
- Strong empirical performance compared with the conventional GP method.
- The paper is well-written and easy to follow, and it is accompanied by good visualization.

**Weaknesses:**

- **Crossover**: Is fitness guidance truly necessary? Could we instead prompt with "improve on the above decision trees" and still achieve comparable performance? Have you conducted any ablation studies on this?

- **Mutation**:
  - Are the log probabilities of within-population solutions consistently higher than those of out-of-population solutions? Have you verified this in your experiments?
  - According to your experiments, LLEGO sacrifices population diversity compared to traditional GP due to its semantic priors. Could you implement mutation alone to preserve population diversity? If not, does this suggest that the designed diversity-guided mutation operator fails to maintain sufficient diversity?

**Questions:**

- Can LLEGO be extended to other string optimization problems, or is it currently too specialized for the Decision Tree Induction benchmarks it addresses?
- What does "LLEGO_no_prior" mean? Have you compared LLEGO to vanilla LLM evolution without semantics guidance in crossover or diversity guidance in mutation?
- Recent work has incorporated semantic priors generated by LLM reflections into the evolutionary process [1]. A discussion of how this approach relates to LLEGO would be beneficial.

[1] ReEvo: Large Language Models as Hyper-Heuristics with Reflective Evolution, NeurIPS 2024

---

> ### Author Response · Authors · 2024-11-21
> **Response to Reviewer BF7z (Part 1/2)**
>
> *We appreciate the reviewer’s detailed and thoughtful evaluation and positive feedback.*
>
> ---
>
>
> ### [P1] Fitness-guidance
>
> Thank you for this suggestion. We would like to point to a few pieces of empirical evidence to support the value of explicit fitness guidance.
>
> 1. **Effect of $\alpha$.** Our analysis in `Fig 5` shows how fitness guidance ($\alpha \in$ {$-0.25, -0.1, 0.1, 0.25$}) enables precise control over the exploration-exploitation tradeoff. Higher $\alpha$ values produce fitter offspring while reducing diversity, allowing a systematic balance of these competing objectives.
> 2. **Ablation study.** Our experiments in `Fig 7` demonstrate that removing fitness-guided crossover (LLEGO$_\mathrm{no xo}$) degrades both search efficiency and generalization performance.
> 3. **Additional ablation.** Following your suggestion, we evaluated a simpler 'improve the solution'-type prompt without fitness guidance. Results across 7 classification tasks (3 seeds each) showed consistently inferior performance compared to our fitness-guided approach (see `App E.2`).
>
> These results collectively demonstrate that explicit fitness guidance through $\alpha$ provides more reliable optimization and superior performance compared to general improvement prompts.
>
> **Actions taken:** We have included additional ablation results in `App E.2`
>
> ---
>
> ### [P2] Mutation operator properties
>
> **Logprobs.** Thank you for this concrete recommendation. We analyzed the relationship between offspring log probabilities and their structural similarity to parent trees. By generating $1000$ offspring through our mutation operator, we computed the Tree-Edit Distance (TED) [1] between each parent-offspring pair. Our analysis in `App E.7` reveals a strong negative correlation ($\rho=-0.85$), indicating that the LLM assigns lower log probabilities to offspring that are more structurally distant from their parents.
>
> [1] Philip Bille. A survey on tree edit distance and related problems. Theoretical computer science, 337 (1-3):217–239, 2005.
>
> **Role of mutation and diversity.** Thank you for this important observation. Please allow us to clarify several key points:
>
> **Semantic priors and lower diversity.** The reduced diversity in LLEGO primarily stems from its use of semantic priors. While traditional GP places uniform prior over all solutions permitted by variation operators, LLEGO's operators concentrate probability mass on semantically meaningful solutions. This naturally focuses search on smaller but more promising regions of the solution space compared to GATree (manifesting as lower diversity metrics)—an effect also observed in traditional semantic GP approaches [2].
>
> **Lower vs insufficient diversity.** The distinction between *lower* and *insufficient* diversity is crucial:
> * While LLEGO shows lower diversity compared to traditional GP, as highlighted above, this is evidence for the guidance of semantic priors.
> * Our empirical results demonstrate that this semantically guided exploration leads to a significantly more efficient search than GATree, suggesting the diversity level is indeed sufficient.
> * Our ablation study (`Fig 7`) systematically investigates the removal of fitness-guided crossover, diversity-guided mutation, and semantic priors. The results show these components work in concert to achieve optimal search efficiency, with mutation specifically helping maintain necessary diversity levels while benefiting from semantic guidance.
>
> We believe these results demonstrate that LLEGO achieves an effective balance between exploration and exploitation through its complementary operators, despite lower absolute diversity levels.
>
> [2] Krzysztof Krawiec and Tomasz Pawlak. Approximating geometric crossover by semantic backpropagation. In Proceedings of the 15th annual conference on Genetic and evolutionary computation, pp. 941–948, 2013.
>
> ---
>
> ### [P3] Extension to other domains
>
> We thank the reviewer for this fruitful suggestion. To facilitate future research building on our framework, we have included a general 'recipe' for extending LLEGO to other domains, such as symbolic regression or program synthesis.

---

> ### Author Response · Authors · 2024-11-21
> **Response to Reviewer BF7z (Part 2/2)**
>
> **General recipe.** We provide a detailed discussion of this recipe and its implementation considerations in `App E.8`, which we summarize here in brief. Using symbolic regression as an illustrative example, adapting LLEGO would require the following components:
> 1. **Representation:** of mathematical expression in natural language that LLMs can interpret and manipulate (e.g. standard notation)
> 2. **Offspring validation/parsing:** implementing validation mechanisms to confirm LLM-evolved equations are mathematically valid, and parsing them back into machine-executable formats (e.g. using SymPy)
> 3. **Fitness function definition:** defining appropriate fitness metrics such as mean-squared error, or equation complexity
> 4. **Domain-specific prompts:** incorporating domain knowledge about function properties (e.g. monotonicity, periodicity) to guide search
>
> This extensibility demonstrates LLEGO's potential as a general framework for LLM-guided optimization beyond decision tree induction.
>
> **Actions taken:** We have included a detailed adaptation recipe in `App E.8`.
>
> ---
>
>
> ###  Questions
>
> **No prior ablation.** In this ablation, we remove all semantic information from prompts (see `Listings 3-4` for concrete examples). Specifically, we replace semantic feature names with generic identifiers ($X_i$) and remove task descriptions. Results in `Sec 5.3` show this ablation achieved worse search performance, demonstrating the additional performance gains of incorporating semantic priors in guiding optimization.
>
> **Relationship to [3]**: Thank you for bringing this relevant work to our attention. While both works leverage LLMs in an evolutionary process, they differ in fundamental ways:
> * **Problem scope:** [3] searches for general meta-heuristics applicable across a set of predefined tasks, whereas LLEGO is specifically designed for decision tree induction, leveraging dataset-specific characteristics and domain knowledge.
> * **Search methodology:** [3] relies on language instruction prompts to generate variations, while LLEGO introduces different mechanisms: fitness-guided crossover through conditioning on $f^*$ and diversity-guided mutation via $\tau$-controlled sampling (of computed logprobs). These mechanisms enable systematic control over exploitation and exploration at the population level.
> * **Future work:** Incorporating reflection mechanisms from [3] could potentially enhance LLEGO's performance and represent an interesting direction for future research.
>
> **Action taken**: We have included a discussion on [3] at `L492` and `L874` in the revised manuscript.
>
> [3] Haoran Ye, Jiarui Wang, Zhiguang Cao, and Guojie Song. Reevo: Large language models as hyper-heuristics with reflective evolution. NeurIPS, 2024
>
>
> ---
>
> *We hope the reviewer’s concerns are addressed and they will consider updating their score. We welcome further discussions.*

---

> > ### Comment · Reviewer_BF7z · 2024-11-25
> > **Thanks for your rebuttal**
> >
> > Thank you for your detailed response, which effectively addressed my concerns.
> >
> > However, I urge the authors to further elaborate on the questions raised by Reviewer mCct regarding optimization and generalization.
> >
> > Additionally, I encourage the authors to discuss whether data contamination in LLM training might impact the reliability of your evaluation results, considering your use of 12 long-established open-source datasets.

---

> > > ### Author Response · Authors · 2024-11-25
> > > **Follow-up Response**
> > >
> > > We are pleased that our response addressed your concerns. Regarding optimization and generalization, we refer to our detailed response to Reviewer mCct's comment.
> > >
> > > On the question of memorization, we provide several pieces of evidence investigating these concerns:
> > > 1. In `App D.2` (especially `Table 8`), we evaluated LLEGO on proprietary datasets (requiring authorized access, and hence highly unlikely to be in the LLM training corpus), demonstrating consistently superior performance.
> > > 2. Our convergence analysis (`Fig 4`), describes systematic improvement across generations--inconsistent with memorization, which would manifest as near-optimal solutions in early generations.
> > >
> > > ---
> > >
> > > *We hope the reviewer’s concerns are addressed and they will consider updating their score. We welcome further discussions.*

---

> > > > ### Comment · Reviewer_BF7z · 2024-11-26
> > > > **Thanks for your follow-up response**
> > > >
> > > > All of my concerns have been addressed. I believe this work demonstrates a solid application of LLM+GP to Decision Tree Induction, and I am willing to recommend its acceptance.
> > > >
> > > > Since the methodology overall (LLM+GP) is not particularly novel, I maintain my score, which is above the acceptance threshold.

---

> > > > > ### Author Response · Authors · 2024-11-28
> > > > > **Thank you**
> > > > >
> > > > > Dear Reviewer **BF7z**,
> > > > >
> > > > > We sincerely appreciate your thorough engagement with our work and the concrete feedback that helped strengthen the paper. We are pleased that our responses addressed your comments and led to your recommendation for acceptance.
> > > > >
> > > > > With thanks,
> > > > >
> > > > > The Authors of #12082

---

### Official Review · Reviewer_mCct · 2024-10-31

**Soundness:** 3
**Presentation:** 2
**Contribution:** 2
**Rating:** 5
**Confidence:** 4

**Summary:**

In this paper, the authors propose using LLMs as evolutionary operators to evolve decision trees.

**Strengths:**

The strength of this paper is that it explores decision tree induction in the era of large language models and shows promising results compared to traditional evolutionary algorithms and optimal decision tree induction methods.

**Weaknesses:**

The weakness of this paper is that the motivation for using LLMs for decision tree induction is not well justified. It is unclear why LLMs would be helpful for decision tree induction, especially considering that LLMs can still function without semantic priors, as shown in Appendix D.2.

**Questions:**

Here are several questions that need to be addressed:
1. The most important issue in this paper is the claim that existing optimal decision tree methods are computationally complex, with complexity scaling exponentially with depth and the number of possible splits in the dataset. However, in the experimental results, the authors show their method with depths of 3 and 4, which can actually be solved by optimal decision tree induction methods, such as DL8.5 [1]. As shown in the DL8.5 paper, with a depth limit of 4, DL8.5 can find the optimal decision tree in 19 out of 24 cases within 10 minutes. It is unclear why the authors chose to use LLMs, an expensive method, to address this issue.

2. The idea of using large language models for evolving programs with diversity considerations is not novel. For example, the GECCO 2024 paper "LLMatic: Neural Architecture Search via Large Language Models and Quality Diversity Optimization" [2] already considers this. Also, using random selection for parent selection in diversity-guided mutation is insufficient. Please consider quality-diversity optimization for parent selection.

3. Please report the training accuracy as well. It is hard to understand why the proposed method would outperform optimal decision trees within depths of 3 and 4.

4. The proposed method is tested on GPT-3.5. Are the conclusions applicable to other models, such as GPT-4 or Claude? Please conduct experiments to verify this. Analyzing the sensitivity of the proposed method to different LLMs is also important.

5. The paper claims one advantage of the proposed method is using LLMs to apply semantic information to guide the search. However, the conclusion of Table 7 shows that LLMs work well even when semantics are removed. The authors should reconsider why LLMs perform well in this context.

6. The term "semantic information" seems more like "domain knowledge" rather than the commonly understood meaning of semantics in the genetic programming domain. Please consider changing the term or at least add an explanation to clarify the difference. Otherwise, it could be confusing.

References:

[1]. Aglin, Gaël, Siegfried Nijssen, and Pierre Schaus. "Learning optimal decision trees using caching branch-and-bound search." Proceedings of the AAAI Conference on Artificial Intelligence. Vol. 34. No. 04. 2020.

[2]. Nasir, Muhammad Umair, et al. "LLMatic: Neural Architecture Search via Large Language Models and Quality Diversity Optimization." Proceedings of the Genetic and Evolutionary Computation Conference. 2024.

---

> ### Author Response · Authors · 2024-11-21
> **Response to Reviewer mCct (Part 1/2)**
>
> *We appreciate the reviewer’s detailed and thoughtful evaluation and feedback.*
>
> ---
>
> ### [P1] Comparison against optimal induction methods
>
> Thank you for raising this concern. Allow us to clarify a few points.
>
> **Computational complexity.** While DL8.5 and related optimal induction methods can find solutions within reasonable time constraints, their applicability is mainly limited to (1) classification problems, with (2) a smaller number of possible splits, and (3) lower depths. The depth=$4$ results referenced from the DL8.5 paper (Table 2) investigate particular datasets with binarized features and relatively fewer possible splits (see 'nItems' column). Indeed, the authors of DL8.5 observed that optimization times-out at depth $\geq 3$ on datasets with continuous features and higher numbers of unique values (Table 3).
>
> **Broader applicability.** Additionally, we highlight that LLEGO's contributions extend beyond addressing computational complexity. As shown in `Table 3` (`App A`), our method is more broadly applicable. In addition to scaling to deeper trees and larger datasets, LLEGO can:
> 1. Handle regression tasks effectively, where we demonstrate consistent performance improvements across multiple regression tasks (`Table 2`)
> 2. Optimize trees for arbitrary objective functions, including fairness-aware objectives (`App D1`). Here, LLEGO obtained a set of Pareto-efficient solutions trading off accuracy and fairness, whereas optimal induction methods are restricted to classification objectives
>
> Your feedback has helped us realize we didn't prominently highlight these advantages, and we have revised the main paper accordingly.
>
> **Generalization performance.** Our empirical results demonstrate that LLEGO consistently outperforms optimal induction methods on generalization performance, underscoring benefits beyond computational efficiency. **Additional results.** To further demonstrate LLEGO's generalization capabilities, we include additional results for trees with depth $5$ in `App E.1`, where we observe consistent performance gains.
>
> **Value proposition.** While we acknowledge that LLEGO has higher computational requirements, we believe this trade-off is justified where optimal performance is crucial (e.g. in safety-critical domains), or for problems with continuous, multi-objective objectives (e.g. balancing accuracy and fairness). A promising direction for future work lies in reducing computational overhead while maintaining LLEGO's performance advantages.
>
> **Actions taken:** We have (1) revised `L112-115`, `L313-314` to clarify the broader applicability of LLEGO, (2) introduced additional results on depth=$5$ problems in `App E.1`.
>
> ---
>
> ### [P2] Related work
>
> Thank you for bringing this relevant work to our attention.
>
> LLMatic introduces a novel technique integrating LLMs and quality-diversity (QD) framework to perform neural architecture search (NAS), maintaining dual archives to evolve both neural architectures and variation prompts. While LLMatic's findings align with our observation that LLM-based optimization achieves superior search efficiency (in their respective domains), there are significant differences between our approaches:
> 1. **Novel mechanisms:** LLEGO introduces novel mechanisms, namely fitness-guided crossover through conditioning on $f^*$ and diversity-guided mutation via temperature-controlled ($\tau$) sampling based on normalized offspring log probabilities. This contrasts with LLMatic's approach of using behavioral descriptors and archive-based QD optimization
> 2. **Prompt design:** LLEGO leverages in-context learning to achieve fitness conditioning (for exploitation) and to compute comparable logprobs to guide exploration. In contrast, LLMatic relies on natural language instructions to generate variations (e.g. *'improve the above network'*)
> 3. **Different applications:** LLEGO demonstrates the effectiveness of LLM-based optimization for decision tree induction, achieving strong performance on this traditional problem that has not previously been considered using natural language representations, while LLMatic explores applications in NAS
>
> **QD parent sampling.** We agree that our framework can be adapted to accommodate different parent sampling schemes. In LLEGO's current form, we utilized random sampling, as the main focus of our mutation operator was the use of computed offspring logprobs to guide diversity. We performed a preliminary investigation based on your interesting suggestion, which we detail in `App E.5` and summarize briefly here.

---

> > ### Comment · Reviewer_mCct · 2024-11-21
> >
> > Thank the authors' response. My main concerns now focus on P1 and P3. These questions are essentially asking the same thing: whether LLMs are genuinely suitable for decision tree induction.
> >
> > First, based on the response to P3 and the newly added experimental results, we can see that LLEGO is inferior to DL8.5 in terms of training performance. From an optimization perspective, this suggests that LLEGO is actually less effective than DL8.5 for solving the decision tree induction problem.
> >
> > Based on the response to P3, now looking at P1, the response mentions that LLEGO demonstrates better generalization performance. However, it is unclear why LLEGO achieves this. One critical point to consider is the evaluation metric—balanced accuracy. DL8.5 actually supports setting sample weights. Therefore, please configure sample weights to ensure that DL8.5 is optimized for balanced accuracy.
> >
> > Additionally, there might be a potential issue of data leakage related to the prompts. For example, the label distribution and range should be derived from the training data rather than the entire dataset. In real-world applications, the label distribution and range for unseen data are typically unknown. Ideally, these should be based only on the training data.

---

> ### Author Response · Authors · 2024-11-21
> **Response to Reviewer mCct (Part 2/2)**
>
> We used CVT-MAP-Elites (the algorithm used in LLMatic) for parent sampling in mutation operations. Specifically, we used each parent's functional signature $h(t) \in \mathcal{H} \subseteq \mathbb{R}^d$ as behavioral descriptors, with membership determined by $M$ centroids fitted in $\mathcal{H}$. We observed that varying the number of niches $M \in$ {$1, 5, 10, 20$} can effectively control offspring diversity, suggesting a promising future integration of such techniques with LLEGO's temperature-based diversity guidance.
>
> **Actions taken:** We have incorporated discussions on LLMatic in `L341` and `L869`, and included additional results on QD parent sampling in `App E.5`.
>
> ---
>
> ### [P3] Training set performance
>
> We appreciate this concrete recommendation. We have added training set performance for all baselines in `Table 17` (`App E.3`). As expected, DL8.5, being an optimal induction approach, achieves the highest training accuracy. However, our analysis reveals that while optimal methods excel in training, LLEGO demonstrates superior generalization. This difference becomes more pronounced at greater tree depths ($3 \rightarrow 5$), where the risk of overfitting increases.
>
> LLEGO's better generalization can be attributed to its use of semantic priors during search—a finding consistent with observations in LLMatic. This may explain why LLEGO outperforms optimal methods in generalization: their focus on training performance can lead to learning spurious patterns. While the generalization capabilities of optimal tree induction methods remain an active research question, our findings are consistent with recent empirical studies [1, 2, 3].
>
>
> [1] Sullivan, C., Tiwari, M. and Thrun, S., 2024, March. MAPTree: Beating “Optimal” Decision Trees with Bayesian Decision Trees. In Proceedings of the AAAI Conference on Artificial Intelligence (Vol. 38, No. 8, pp. 9019-9026).
>
> [2] Zharmagambetov, A., Hada, S.S., Gabidolla, M. and Carreira-Perpinán, M.A., 2021, July. Non-greedy algorithms for decision tree optimization: An experimental comparison. In 2021 International Joint Conference on Neural Networks (IJCNN) (pp. 1-8). IEEE.
>
> [3] Marton, S., Lüdtke, S., Bartelt, C. and Stuckenschmidt, H., 2024, March. GradTree: Learning axis-aligned decision trees with gradient descent. In Proceedings of the AAAI Conference on Artificial Intelligence (Vol. 38, No. 13, pp. 14323-14331).
>
> ---
>
> ### [P4] Evaluating different LLMs
>
> Thank you for this concrete recommendation. LLEGO's design is LLM-agnostic, and our additional experiments with GPT-4 (`Table 18`, `App E.4`) show performance improvements across most tasks. These results demonstrate that LLEGO is not only robust across LLM architectures but can also naturally scale with advances in underlying LLMs.
>
> ---
>
> ### [P5] Importance of semantic priors
>
> We appreciate this insightful observation. While LLEGO performs even without semantic information (LLEGO$_\mathrm{noprior}$), this demonstrates the synergy between two key aspects of our method:
> 1. First, the LLM provides powerful inherent capabilities, including strong pattern recognition from in-context examples, and integrating semantic understanding (when available) to generate semantically meaningful offspring.
> 2. Second, these capabilities are enhanced by LLEGO's framework design, which includes fitness- and diversity-guidance and higher-arity operations.
>
> For intuition as to why LLEGO performs well, even when semantic priors are absent, consider our ablation prompts in `Listings 3-4`. The LLM can leverage pattern recognition to identify performant subtrees (e.g. rooted at 'X_3 < 50') present across parents. However, when semantics are available (e.g. 'age < 50'), the LLM can incorporate additional domain knowledge to enhance search performance.
>
> Empirically, we observed that semantic information provided additional performance gains on 8/10 tasks, yielded more consistent results (lower standard deviations), and enabled more efficient search (`Fig 7`).
>
> We view this flexibility as a core strength: LLEGO works effectively in domains with limited semantic information but can leverage available semantics for additional gains, making it broadly applicable.
>
> ---
>
> ### [P6] Clarifying terminology
>
> Thank you for noting this distinction. In semantic GP, semantics typically refers to a program's functional signature, while we use the term to describe domain knowledge about features and the solution space encoded in the LLM. We have clarified this distinction in `L163`.
>
>
> ---
>
> *We hope the reviewer’s concerns are addressed and they will consider updating their score. We welcome further discussions.*

---

> ### Author Response · Authors · 2024-11-22
> **Follow-up Response (1/2)**
>
> *We appreciate your thoughtful and prompt follow-up and are pleased to have addressed many of your concerns.*
>
> ---
>
> With regards to your latest comments, while DL8.5 provides theoretical guarantees of optimality in *select settings* on the *training set*, this framing overlooks several crucial considerations:
>
> 1. **Types of objectives.** DL8.5's guarantee of global optimality is restricted to a specific class of objective functions (Eq 1 in the DL8.5 paper) that applies only to classification tasks. This precludes: (1) regression objectives; (2) complex multi-objective or regularized objectives (e.g. fairness).
>
> Essentially, DL8.5 optimizes a specific objective (Eq 1, which can be adjusted with `sample_weights`), but this is fundamentally different from LLEGO which can directly optimize arbitrary objectives or metrics of interest (e.g. balanced accuracy). To this end, we also demonstrated LLEGO's efficacy on regression objectives (`Table 2`) and fairness-regularized objectives (`App D1`).
>
> 2. **Training vs generalization performance.** We feel that your comment *'from an optimization perspective... LLEGO is actually less effective than DL8.5 for solving decision tree induction'* overlooks a crucial nuance. We emphasize that training set performance is not the primary goal. Indeed, any arbitrarily deep tree can, in principle, achieve perfect training accuracy. The true measure of efficacy lies in **generalization performance**, where LLEGO consistently demonstrates advantages. Similar to how L1/L2 regularization sacrifices training performance for better generalization, LLEGO's semantic priors act as a form of implicit regularization derived from domain knowledge to enhance generalization.
>
> For a concrete example of this, consider our new analysis (attached below). We observe that even though all methods achieve strong training performance on **heart** and **liver** datasets, only LLEGO maintained robust generalization on the test set.
>
> 3. **Practical considerations.** DL8.5's exponential complexity with respect to unique feature values and tree depth limits its practical utility. In contrast, the computational complexity of LLEGO (and the broader GP family) is independent of these problem-dependent characteristics, making it more viable for many real-world applications.
>
> Succinctly summarized, LLEGO offers three key advantages: (1) broader applicability to a wider class of problems (and objectives), (2) superior generalization performance through semantic priors, and (3) practical viability for problems beyond the computational reach of optimal methods.
>
> ---
>
> ### Additional results
>
>
> * **Sample weights in DL8.5:** Following your suggestion, we conducted additional experiments with DL8.5 incorporating sample weights, maintaining the experimental protocol described in Sec 5.1. While this modification improved performance on some datasets, the overall impact was inconclusive. Notably, LLEGO continues to demonstrate superior performance compared to both weighted and unweighted variants of DL8.5.
>
> * **Data processing clarification:** Regarding the concern about potential data leakage, we confirm that all summary statistics (including label distributions and feature ranges) are computed exclusively from the training split.
>
> ---
>
> *We thank the reviewer for your help in improving our work. Please let us know if our latest responses have addressed your concerns and if there is anything else you would like to see.*

---

> > ### Author Response · Authors · 2024-11-22
> > **Follow-up Response (2/2)**
> >
> > * `Depth=3`, `training` set, balanced accuracy
> >
> > | Method | Breast | Compas | Credit | Diabetes | Heart | Liver | Vehicle |
> > |---|:---:|:---:|:---:|:---:|:---:|:---:|:---:|
> > | DL8.5 (old) | $0.990_{( 0.007) }$ | ${0.692_{( 0.005) }}$ | $0.711_{( 0.021) }$ | ${0.803_{( 0.013) }}$ | ${0.927_{( 0.026) }}$ | ${0.776_{( 0.026) }}$ | ${0.984_{( 0.002) }}$ |
> > | DL8.5 (new) | $0.992_{( 0.007) }$ | $0.692_{( 0.005) }$ | $0.700_{( 0.091) }$ | $0.827_{( 0.016) }$ | $0.925_{( 0.029) }$ | $0.774_{( 0.031) }$ | $0.983_{( 0.003) }$ |
> > | LLEGO | $0.981_{(0.007)}$ | $0.675_{(0.006)}$ | $0.713_{(0.016)}$ | $0.784_{(0.017)}$ | $0.871_{(0.023)}$ | $0.732_{(0.018)}$ | $0.956_{(0.009)}$ |
> >
> > * `Depth=3`, `test` set, balanced accuracy
> >
> > | Method | Breast | Compas | Credit | Diabetes | Heart | Liver | Vehicle |
> > |---|:---:|:---:|:---:|:---:|:---:|:---:|:---:|
> > | DL8.5 (old) | $0.944_{(0.006)}$ | ${0.666_{(0.006)}}$ | $0.591_{(0.010)}$ | $0.655_{(0.018)}$ | $0.704_{( 0.040) }$ | $0.565_{(0.028)}$ | $0.938_{(0.016)}$ |
> > | DL8.5 (new) | $0.947_{( 0.007) }$ | $0.665_{( 0.005) }$ | $0.590_{( 0.041) }$ | $0.703_{( 0.024) }$ | $0.688_{( 0.022) }$ | $0.598_{( 0.031) }$ | $0.932_{( 0.008) }$ |
> > | LLEGO | ${0.946_{(0.010)}}$ | $0.652_{(0.004)}$ | $0.677_{(0.004)}$ | ${0.713_{(0.013)}}$ | ${0.736_{(0.021)}}$ | ${0.672_{(0.017)}}$ | ${0.937_{(0.015)}}$ |
> >
> > * `Depth=4`, `training` set, balanced accuracy
> >
> > | Method | Breast | Compas | Credit | Diabetes | Heart | Liver | Vehicle |
> > |---|:---:|:---:|:---:|:---:|:---:|:---:|:---:|
> > | DL8.5 | ${0.998_{( 0.003) }}$ | ${0.705_{( 0.004) }}$ | ${0.789_{( 0.011) }}$ | ${0.827_{( 0.035) }}$ | ${0.951_{( 0.041) }}$ | ${0.836_{( 0.027) }}$ | ${0.990_{( 0.006) }}$ |
> > | DL8.5 (new) | $0.998_{( 0.003) }$ | $0.706_{( 0.005) }$ | $0.704_{( 0.015) }$ | $0.836_{( 0.024) }$ | $0.956_{( 0.035) }$ | $0.816_{( 0.034) }$ | $0.953_{( 0.060) }$ |
> > | LLEGO | $0.984_{(0.005)}$ | $0.678_{(0.007)}$ | $0.722_{(0.016)}$ | $0.807_{(0.003)}$ | $0.894_{(0.020)}$ | $0.761_{(0.028)}$ | $0.954_{(0.014)}$ |
> >
> > * `Depth=4`, `test` set, balanced accuracy
> >
> > | Method | Breast | Compas | Credit | Diabetes | Heart | Liver | Vehicle |
> > |---|:---:|:---:|:---:|:---:|:---:|:---:|:---:|
> > | DL8.5 (old) | $0.941_{(0.011)}$ | ${0.662_{(0.004)}}$ | $0.586_{(0.015)}$ | $0.636_{(0.025)}$ | $0.744_{( 0.037) }$ | $0.588_{(0.023)}$ | $0.931_{(0.009)}$ |
> > | DL8.5 (new) | $0.939_{( 0.011) }$ | $0.661_{( 0.004) }$ | $0.576_{( 0.015) }$ | $0.671_{( 0.018) }$ | $0.706_{( 0.052) }$ | $0.561_{( 0.017) }$ | $0.898_{( 0.058) }$ |
> > | LLEGO | ${0.951_{(0.006)}}$ | ${0.662_{(0.003)}}$ | ${0.684_{(0.009)}}$ | ${0.731_{(0.004)}}$ | ${0.751_{(0.037)}}$ | ${0.676_{(0.019)}}$ | ${0.937_{(0.013)}}$ |

---

> > ### Comment · Reviewer_mCct · 2024-11-22
> >
> > Thanks to the authors' response. I still have two concerns:
> >
> > 1. **From an optimization perspective:**
> >    My major concern is that the proposed method may not be an effective way to use LLMs for inducing decision trees. In the existing literature, decision tree induction has been formulated as a mixed-integer programming (MIP) problem [1] or a constraint programming task [2]. This paper appears to claim that LLMs are superior to MIP solvers for solving combinatorial optimization problems. However, based on the current results, this claim is not convincing.
> >
> >    There are two potential approaches to demonstrate that an LLM-based solver is better than a MIP solver:
> >    - For small-scale problems: Show that LLMs can achieve optimal solutions faster than MIP solvers.
> >    - For large-scale problems: Show that LLMs can achieve reasonably good performance. Currently, the depth limit is set to 4, which does not qualify as large-scale.
> >
> >    Based on the current results, there is no evidence that LLMs can either solve optimal decision tree induction faster than MIP solvers or perform well on large-scale problems.
> >
> >    This issue is separate from generalization performance. As demonstrated by the DL8.5 paper, the focus is purely on optimization performance; the data was not split into training and test sets, as the task was fundamentally about optimization.
> >
> > 2. **From a generalization perspective:**
> >    It is unclear where the implicit regularization comes from. For example, the authors claim that the semantic prior acts as an implicit regularizer. Could the authors show the training performance without the semantic prior to demonstrate that it indeed functions as an implicit regularizer? Additionally, could the authors conduct more ablation studies to clarify the source of implicit regularization? From Table 7, it appears that even without using the prior, LLEGO still achieves good generalization performance.
> >
> > [1]. Aglin, G., Nijssen, S., & Schaus, P. (2020, April). Learning optimal decision trees using caching branch-and-bound search. In Proceedings of the AAAI conference on artificial intelligence (Vol. 34, No. 04, pp. 3146-3153).
> >
> > [2]. Verhaeghe, H., Nijssen, S., Pesant, G., Quimper, C. G., & Schaus, P. (2020). Learning optimal decision trees using constraint programming. Constraints, 25, 226-250.

---

> ### Author Response · Authors · 2024-11-25
> **Follow-up Response (1/2)**
>
> *Thank you again for your continuous engagement and prompt response. We address your comments below.*
>
> ---
>
> ### From an optimization perspective
>
> **Addressing the MIP/CP comparison.** We understand your concerns, but we respectfully suggest that comparing LLEGO to exact MIP/CP solvers **mischaracterizes** its core contribution and purpose. While MIP/CP solvers guarantee optimal solutions on the training set when tractable, LLEGO addresses fundamental ML goals: learning trees that generalize well to unseen data, is more broadly applicable to different types of problems, and practically viable for larger problems.
>
> |  | Optimal induction (MIP/CP solvers) | LLEGO |
> |---|---|---|
> | Generalization | Potential overfitting to training set, reducing generalization performance | Utilizes semantic priors to discover more generalizable solutions |
> | Applicability | Limited to specific types of classification objectives | Applicable to arbitrary objective functions (including regression, multi-objective) |
> | Practical viability | Times out for larger problems (dataset and tree depth-dependent) | Constant complexity with respect to dataset characteristics and tree-depth |
> | Evaluation metric | Optimality gap and runtime  | Generalization performance|
>
>
>  Regarding the statement
> > 'This paper appears to claim that LLMs are superior to MIP solvers for solving combinatorial optimization problems.'
>
> We want to clarify that **this is not a claim we make**. Our focus has consistently been on generalization (and not optimization on training set), broader applicability, and practical viability. This focus on generalization was explicitly stated in our abstract (`L21`) and prior responses. If any part of our manuscript inadvertently suggests otherwise, we welcome specific feedback for revision.
>
> Lastly, we wish to emphasize that decision tree induction algorithms should fundamentally be evaluated on their ability to generalize from limited observations. (i.e. **induction**). This aligns with both real-world considerations and standard machine learning practices [1, 2]. Along this front, our empirical analysis highlights the generalization strength of LLEGO, outperforming all baselines in $19/21$ problem instances across trees of depths $3-5$.
>
>
> [1] Costa, V.G. and Pedreira, C.E., 2023. Recent advances in decision trees: An updated survey. Artificial Intelligence Review, 56(5), pp.4765-4800.
>
> [2] Barros, R.C., Basgalupp, M.P., De Carvalho, A.C. and Freitas, A.A., 2011. A survey of evolutionary algorithms for decision-tree induction. IEEE Transactions on Systems, Man, and Cybernetics, Part C (Applications and Reviews), 42(3), pp.291-312.
>
>
> ---
>
> ### Sources of semantic prior
>
> Thank you for this question. Let us consider LLEGO vs traditional genetic programming (GP) to understand this semantic prior.
> * **Traditional GP**: The space of possible offspring is defined by the chosen genetic operator, typically with a uniform prior over possible offspring. The only 'inductive bias' comes through operator design and permitted operations.
> * **LLEGO**: By using LLMs as genetic operators, LLEGO inherits semantic priors from large-scale pretraining, leading to a non-uniform distribution that favors meaningful tree structures.
>
> **Empirical support.** This is investigated systematically in our paper through three key findings:
> 1. For XO operators, controlling $f^*$ reliably altered the fitness/diversity characteristics of evolved offspring (`Fig 5`)
> 2. For MUT operators, computed offspring logprobs correlate with structural distance (`Fig 22`), allowing us to reliably control mutation diversity (`Fig 6`)
> 3. Our analysis of search dynamics (`Fig 4`) demonstrated LLEGO's significantly higher search efficiency compared to traditional GP (no semantic priors), providing strong evidence for the effectiveness of LLM-derived semantic priors
>
> **Theoretical support.** This effect aligns with theoretical work by [1], which demonstrates that in-context learning performs implicit Bayesian inference by marginalizing over pretraining concepts contained in a language model.
>
>
>
> [1] Xie, S.M., Raghunathan, A., Liang, P. and Ma, T., An Explanation of In-context Learning as Implicit Bayesian Inference. In International Conference on Learning Representations.

---

> ### Author Response · Authors · 2024-11-25
> **Follow-up Response (2/2)**
>
> **Additional results.** To provide a concrete demonstration of how semantic priors affect generalization, we conducted a controlled experiment with spurious correlations. While simplified, this experiment offers clear insights into the regularizing effects of semantic priors. We introduced a spurious feature (named 'random_feature') that perfectly predicts labels on $50\\%$ of training/validation samples but is uninformative at test time. Purely data-driven approaches (DL8.5, LLEGO$_{no\\_prior}$) incorporated this spurious feature into their solutions [as shown here](https://imgur.com/a/FmEesvs), achieving higher training accuracy but worse generalization. In contrast, LLEGO, by leveraging semantic priors to assess feature relevance, avoided these spurious correlations entirely. This provides direct evidence that LLM-derived semantic priors serve as an implicit regularizer, improving generalization performance.
>
>
> | Model | Training | Test |
> |--------|-----------|------|
> | DL85 | $0.809 \pm 0.008$ | $0.569 \pm 0.065$ |
> | LLEGO$_{no\\_prior}$ | $0.704 \pm 0.017$ | $0.641 \pm 0.019$ |
> | LLEGO | $0.676 \pm 0.016$ | $0.682 \pm 0.015$ |
>
>
> These additional results, combined with existing analysis, demonstrate that LLM-derived semantic priors enhance both search efficiency and generalization. We acknowledge that characterizing their precise mechanisms and properties remains an important direction for a dedicated future work.
>
>
> ---
>
> *Thanks again, we appreciate your thorough engagement with our work. If you have more questions or concerns, please let us know.*

---

> > ### Comment · Reviewer_mCct · 2024-11-26
> >
> > Thank you for the further analysis. Let’s now focus on the generalization issue. I have four questions:
> >
> > 1. **The example in additional results is not supervised learning.**
> > The new example is not a valid instance of supervised learning because it violates the i.i.d. assumption. The spurious features are informative during training, achieving an accuracy of around 75%. However, these features act as random guesses in the test data, indicating that the distribution $P(Y|X)$ has changed. In such cases, prior knowledge is obviously helpful. However, performing feature selection based on domain knowledge could achieve the same effect.
> >
> > 2. **Concern about using 40% of the data to tune hyperparameters of DL8.5.**
> > If I understand correctly, the baseline algorithm DL8.5 is trained using 20% of the training data and 40% for hyperparameter tuning. In a typical machine learning pipeline, hyperparameters are tuned through cross-validation, allowing 60% of the data to be used for training DL8.5. Please consider training DL8.5 with this setup.
> >
> > 3. **Semantic prior does not seem to act as regularization.**
> > From my observation of Figure 15, after applying the semantic prior, LLEGO achieves better fitness on the training set. This suggests that the semantic prior is used to improve training accuracy rather than serving as implicit regularization to control overfitting.
> >
> > 4. In LLEGO, validation loss is used to select the final decision tree from the search history. Please clarify whether this strategy is also applied in GATree.

---

> ### Author Response · Authors · 2024-11-28
> **Author's Response**
>
> *We thank the reviewer for their thorough and constructive engagement during the discussion period. We address the key points in your latest follow-up below.*
>
> ---
>
> ### Response to Q1
>
> We emphasize that our additional experiment specifically investigates **out-of-distribution (OOD) generalization**, following standard OOD experimental protocols, where spurious correlations are present in train/val splits but absent in test splits [1, 2].
>
> The main purpose of this experiment is to provide easily **analyzable insights** into LLEGO's superior generalization performance through a concrete example of semantic prior incorporation. The core set of **in-distribution (ID) generalization** results are those already reported in the manuscript, where LLEGO outperforms baselines on 19/21 problems.
>
> Additionally, we make the following notes:
> 1. Traditional data-driven feature selection approaches (possibly performed using a validation set) are challenging to apply here, as the spurious correlation exists in both train and validation splits, only revealing its spurious nature in test data.
> 2. While domain knowledge-guided feature selection could help, this would inherently leverage **semantic priors**—precisely what LLEGO achieves systematically through its semantic-aware operators.
>
> [1] Beery, S., Van Horn, G., & Perona, P. (2018). Recognition in terra incognita.
>
> [2] Liu, J., Shen, Z., He, Y., Zhang, X., Xu, R., Yu, H., & Cui, P. (2021). Towards out-of-distribution generalization: A survey.
>
> ---
> ### Response to Q2
>
> We appreciate this suggestion. We conducted the recommended experiment using $60\\%$ of the dataset for training DL8.5, with hyperparameters selected through cross-validation. To ensure complete convergence on this larger training set, we allocated DL8.5 **2x the computational budget**, compared to the standard budget used by LLEGO and other baselines: 10 minutes for model fitting, in addition to 10 rounds of hyperparameter tuning.
>
> The results are consistent with our original findings: LLEGO maintains **superior generalization performance** even with DL8.5's increased training dataset and computational allocation.
>
>
> | Method | Breast | Compas | Credit | Diabetes | Heart | Liver | Vehicle|
> |--------|---------|---------|---------|-----------|--------|--------|----------|
> DL85 | $0.947 \pm 0.005$ | $0.663 \pm 0.002$ | $0.659 \pm 0.003$ | $0.683 \pm 0.027$ | $0.714 \pm 0.024$ | $0.616 \pm 0.009$ | $0.941 \pm 0.004$ |
> | LLEGO | $0.951 \pm 0.006$ | $0.662 \pm 0.003$ | $0.684 \pm 0.009$ | $0.731 \pm 0.004$ | $0.751 \pm 0.037$ | $0.676 \pm 0.019$ | $0.937 \pm 0.013$ |
>
> ---
>
> ### Response to Q3
>
> We appreciate your observations. Indeed, the relationship between semantic priors and model performance is nuanced and depends on the comparison baseline (DL8.5 or LLEGO$_{no\\_prior}$).
>
> Our [previous characterization](https://openreview.net/forum?id=UyhRtB4hjN&noteId=qdokIIUFfe) of LLEGO's semantic priors as **implicit regularization** was specifically in comparison to DL8.5, not LLEGO$_{no\\_prior}$. This can be better understood by distinguishing between **two types of prior effects** on model performance:
>
> | Effects of priors | Regularizing priors | Informative priors |
> |---|---|---|
> | Mechanism | Limiting model capacity/complexity | Encodes general patterns/domain knowledge |
> | Training performance | Generally decreases | Generally improves |
> | Generalization performance | Generally improves | Generally improves |
> | Examples | L1/L2, depth constraints, avoiding spurious features | Physical laws, image invariances, semantic knowledge (e.g. feature hierarchies and interactions) |
>
> In LLEGO, the semantic prior serves **both roles**, but their dominant effect varies by comparison:
> * Versus DL8.5: it functions primarily as a **regularizing prior** by constraining specialized splits that might be mathematically optimal yet semantically inconsistent. This may reduce training performance compared to DL8.5's purely data-driven global optimization approach, but improve generalization.
> * Versus LLEGO$_{no\\_prior}$: it acts more as an **informative prior**, by leveraging semantic domain knowledge to guide better splits, potentially improving both training and generalization performance.
>
> This analysis sheds light on (1) the potential improvement in training and generalization performance versus LLEGO$_{no\\_prior}$ (`Fig 15`), and (2) our earlier characterization of regularization effects versus DL8.5.
>
> ---
>
> ### Response to Q4
>
> We applied the same validation loss-based selection strategy to select GATree's final decision tree.
>
> ---
>
> *We trust these responses clarify the key points raised in your follow-up. We appreciate your careful consideration of our work.*

---

> > ### Comment · Reviewer_mCct · 2024-11-30
> >
> > **Based on the experimental results, it seems that LLMs can achieve better generalization performance than traditional decision tree induction methods in some cases. I have raised the score to 5, as this paper demonstrates that ChatGPT can induce decision trees with improved generalization performance.**
> >
> > **However, the main concern is that it is still unclear what mechanisms enable LLMs to achieve better generalization performance. This uncertainty may hinder the applicability of the proposed method, as we do not know when this method can be effectively used and when it cannot.**
> >
> > I acknowledge that the authors have conducted experiments on GPT-3.5 and GPT-4 to confirm that the proposed method works on different versions of GPT models. However, without understanding the mechanisms that allow LLMs to induce decision trees with good generalization performance, the proposed method may fail in the future if OpenAI significantly changes its ChatGPT models.
> >
> > The authors have presented an example of spurious correlation. In my understanding, in this example, the authors explicitly informed the LLM that certain features were spurious. If we have such prior knowledge, these features should ideally be eliminated before the machine learning pipeline, rather than relying on the algorithm to address them.
> >
> > **Besides concerns about generalization, I am also not entirely convinced about using LLMs to solve MIP problems like decision tree induction. Performing arithmetic calculations remains a challenging task for LLMs [1].**
> >
> > While I agree that LLMs can perform feature selection to improve generalization performance [2], the current evidence is still not convincing enough to demonstrate that LLMs can effectively split the feature space recursively.
> >
> > [1]. Yuan, Z., Yuan, H., Tan, C., Wang, W., & Huang, S. (2023). How well do large language models perform in arithmetic tasks? arXiv preprint arXiv:2304.02015.
> > [2]. Jeong, D. P., Lipton, Z. C., & Ravikumar, P. (2024). LLM-Select: Feature Selection with Large Language Models. arXiv preprint arXiv:2407.02694.

---

> ### Author Response · Authors · 2024-12-03
> **Thank you and to summarize**
>
> *We appreciate your engagement with our work and are glad that our responses addressed many of your concerns, which led to an increase in rating.*
>
> We would like to respond to several points from your final remarks:
>
> **Methodological innovations.** We respectfully emphasize that the characterization of our work as *'ChatGPT can induce decision trees'* misrepresents its **core innovations**. LLEGO is a novel genetic programming (GP) framework that introduces genetic operators (specifically, fitness-guided crossover and diversity-guided mutation) that leverage LLMs components for their semantic awareness. These methodological innovations significantly outperform naive LLM prompting (`App E2`), demonstrating the value of our framework.
>
> **Mechanisms for enhanced generalization.**  Through carefully controlled experiments, we systematically analyzed LLEGO's mechanisms for evolving superior trees by contrasting our semantic-aware operators with traditional GP's uniform priors:
> - Conditioning with desired fitness (via $\alpha$) demonstrably steers offspring fitness (`Sec 5.2`, `Fig 5`).
> - Logprob-based sampling (via $\tau$) provides systematic control over population diversity (`Sec 5.2`, `Fig 6`).
> - The interplay between both operators and semantic priors measurably improves search efficiency (`Sec 5.3`, `Fig 7`).
>
> These mechanisms translate to substantial **performance gains**: (1) significantly enhanced search efficiency versus GP (`Fig 4`) and (2) superior generalization on **19/21** classification tasks and **6/8** regression tasks across varying depths (`Tables 1-2`).
>
> **Applicability of LLEGO.** We acknowledge the **No Free Lunch** theorem that no algorithm is universally superior. To this end, we provided a detailed discussion on problem characteristics where LLEGO is expected to excel (`App A.2`). Following ML best practices, practitioners can validate LLEGO's applicability to their own problems through standard **model selection** procedures. LLEGO's performance advantages demonstrated across classification, regression, and multi-objective settings establish it as a **valuable addition** to the decision tree learning toolkit.
>
> **Role of semantic priors.** As we highlighted in [our previous response](https://openreview.net/forum?id=UyhRtB4hjN&noteId=NeF0MGzE5J), the spurious correlation experiment serves as a controlled **demonstration** of a specific **instance** of semantic priors guiding the search process—not evidence that feature selection is the primary (or only) mechanism for improved generalization. The semantic priors play a fundamental role in enabling LLEGO's superior performance by informing operators about meaningful feature relationships and splitting patterns (through in-context learning) to more efficiently explore the space of decision trees and identify structures with better generalization properties.
>
> **Distinct from MIP solvers.** As established in [our previous response](https://openreview.net/forum?id=UyhRtB4hjN&noteId=HYzCZWhS0P), LLEGO fundamentally **differs** from MIP solvers in its: 1) goal of superior generalization, 2) broad applicability across problem types, and 3) practical computational viability. Importantly, LLEGO does not *'perform arithmetic calculations'* anywhere but rather considers existing tree structures (containing numerical values) to identify optimal splitting hierarchies. In this regard, our findings align with recent works demonstrating LLMs' capabilities in manipulating numerical values, especially in the context of optimization tasks [1, 2].
>
> Finally, while we welcome healthy skepticism, we believe that our rigorous experimental design—comprising systematic benchmark selection (`App C1`), comprehensive baseline comparisons, and careful ablation studies—provides compelling evidence for LLEGO's capabilities in evolving trees with superior generalization performance.
>
>
> [1] Yang, C., Wang, X., Lu Y., Liu, H., Le, Q., Zhou, D., and Chen X. Large language models as optimizers, ICLR 2024
>
> [2] Zhang, M.R., Desai, N., Bae, J., Lorraine, J. and Ba, J., 2023. Using Large Language Models for Hyperparameter Optimization. arXiv preprint arXiv:2312.04528.
>
> ---
>
> *We thank the reviewer again for their engagement with our work.*

---

### Official Review · Reviewer_71f5 · 2024-11-03

**Soundness:** 3
**Presentation:** 3
**Contribution:** 3
**Rating:** 6
**Confidence:** 4

**Summary:**

The paper proposes an innovative method for decision tree induction using genetic programming (GP) enhanced by semantic knowledge from large language models (LLMs). The key contribution is the LLEGO framework, which introduces semantically-aware genetic operators: fitness-guided crossover and diversity-guided mutation. These operators leverage semantic priors from LLMs to improve search efficiency and decision tree generalisation performance. The method is empirically validated on various tabular datasets, demonstrating superior performance compared to traditional and state-of-the-art tree induction methods.

**Strengths:**

* The integration of LLMs into GP is a significant advancement. Using LLMs to inform genetic operators with semantic priors is a creative and impactful idea, enhancing both search efficiency and the quality of generated decision trees.
* The paper provides a detailed explanation of the LLEGO framework, including how LLMs are utilised and how genetic operators are implemented. The end-to-end algorithm is clearly described, making it easier to understand the overall workflow.
* The paper addresses the limitations of existing decision tree induction methods by offering a scalable and generalisable approach, which could be beneficial across various domains.

**Weaknesses:**

* The effectiveness of LLEGO depends heavily on the semantic priors embedded in the LLM. If the LLM lacks relevant knowledge for a particular domain or problem, the performance might degrade. A discussion on this limitation and possible solutions would be helpful.
* While the empirical results are strong, the paper lacks a rigorous theoretical analysis of why and how the semantic priors from LLMs improve search efficiency. This could strengthen the understanding of the method’s underlying principles.
* The paper focuses on decision tree induction, but it is unclear how well the approach generalises to other GP-based tasks. A brief discussion or some preliminary results in this direction would make the contribution more comprehensive.

**Questions:**

* How does the computational overhead of using LLMs in LLEGO compare to traditional GP approaches in terms of runtime and memory usage? Are there any scenarios where the increased cost might not be justified? Provide a more detailed analysis of the computational cost and scenarios where the trade-off is most beneficial. Consider discussing potential optimisations or the use of smaller LLMs.

* How does the performance of LLEGO change if the underlying LLM lacks semantic priors relevant to a particular problem domain? Are there strategies to mitigate performance degradation in such cases?

* Include a theoretical explanation of how semantic priors impact search efficiency and decision tree quality. This would strengthen the paper’s scientific contributions.

* Do the authors have any preliminary insights or results on how well LLEGO could be applied to other GP-based problems, such as symbolic regression or evolutionary program synthesis? Test how LLEGO might perform in other GP-based problems, such as symbolic regression or program synthesis, to show the method’s broader applicability.

---

> ### Author Response · Authors · 2024-11-21
> **Response to Reviewer 71f5 (Part 1/2)**
>
> *We appreciate the reviewer’s detailed and thoughtful evaluation and positive feedback.*
>
> ---
>
> ### [P1] Importance of semantic priors
>
> Thank you for this insightful observation regarding LLEGO's dependence on semantic priors.
>
> **Empirical investigations.** Our ablation study (`Fig 7` and `Table 7`) evaluated LLEGO$_{\mathrm{no prior}}$, where we replaced meaningful feature names with generic identifiers and removed task descriptions (see `Listings 3-4`). While this variant performed less competitively than LLEGO, its strong overall performance indicates that semantic priors, while beneficial, are not the sole factor behind LLEGO's efficacy. Please allow us to clarify:
>
> **Complementary mechanisms.** LLEGO's superior performance stems from two key aspects:
> 1. First, the LLM provides powerful inherent capabilities, including strong pattern recognition from in-context examples and integrating semantic understanding (when available) to generate semantically meaningful offspring.
> 2. Second, these capabilities are enhanced by LLEGO's framework design, which includes fitness- and diversity-guidance and higher-arity operations.
>
> As such, even when semantic priors are limited, the LLM's pattern recognition abilities augmented with LLEGO's framework design maintain robust performance.
>
> **Potential enhancements.** For problems with limited semantic knowledge, we identified two promising directions: fine-tuning on domain-specific data and incorporating domain knowledge through prompt construction. LLEGO can also benefit from advances in LLM capabilities, as demonstrated in our evaluation with GPT-4 (`App E.4`), where we achieved stronger performance without modifying the framework.
>
> **Action taken**: We have revised `L488` to discuss assumptions on semantic prior and possible enhancement strategies.
>
> ---
>
> ### [P2] Understanding LLEGO's performance
>
> We appreciate this suggestion. While we recognize the value of theoretical analysis, we prioritized systematic empirical investigations to understand LLEGO's performance in our work. Specifically, we conducted targeted experiments to understand different aspects of performance gain:
>
> 1. **Detailed operator analysis** (`Sec 5.2`): Investigated the contributions of fitness guidance (controlled by $\alpha$) and diversity guidance (controlled by $\tau$) to search efficiency. Key findings revealed complementary mechanisms: crossover effectively exploits promising regions while mutation maintains broader exploration, creating a balanced search dynamic.
> 2. **Comprehensive ablation studies** (`Sec 5.3`): Isolated component contributions by systematically removing semantic priors, fitness guidance, diversity guidance, and higher arity operations. Results demonstrated that LLEGO's performance stems from synergistic interaction between LLM capabilities and framework design.
>
> These insights contribute to our understanding of LLEGO's superior generalization performance compared to greedy and optimal induction methods. While theoretical analysis would be valuable for deeper understanding, we believe it warrants dedicated investigation in future work, given the complexity of analyzing LLMs and GP optimization dynamics.

---

> ### Author Response · Authors · 2024-11-21
> **Response to Reviewer 71f5 (Part 2/2)**
>
> ### [P3] Extension to other domains
>
> We thank the reviewer for this fruitful suggestion. To facilitate future research building on our framework, we have included a general 'recipe' for extending LLEGO to other domains, such as symbolic regression or program synthesis.
>
> **General recipe.** We provide a detailed discussion of this recipe and its implementation considerations in `App E.8`, which we summarize here in brief. Using symbolic regression as an illustrative example, adapting LLEGO would require the following components:
> 1. **Representation:** of mathematical expression in natural language that LLMs can interpret and manipulate (e.g. standard notation)
> 2. **Offspring validation/parsing:** implementing validation mechanisms to confirm LLM-evolved equations are mathematically valid, and parsing them back into machine-executable formats (e.g. using SymPy)
> 3. **Fitness function definition:** defining appropriate fitness metrics such as mean-squared error, or equation complexity
> 4. **Domain-specific prompts:** incorporating domain knowledge about function properties (e.g. monotonicity, periodicity) to guide search
>
> This extensibility demonstrates LLEGO's potential as a general framework for LLM-guided optimization beyond decision tree induction.
>
> **Actions taken:** We have included a detailed adaptation recipe in `App E.8`.
>
> ---
>
> ### [P4] Computational runtime and memory
>
> Thank you for this important question regarding computational considerations:
>
> **Runtime and memory analysis.** All results in `Tables 1-2` were obtained under the same $10$ minute runtime budget, which includes hyperparameter tuning time crucial for baseline performance (detailed in `App C2`). As we utilize API queries, LLEGO's current memory footprint is comparable to traditional GP approaches, only maintaining population and evaluation results locally on CPU. However, deploying local LLMs would incur additional GPU memory requirements.
>
> **Specific comparisons against GATREE.** In `Tables 9-10` (`App D3`), we extended comparisons by significantly increasing GATree's search budget ($N=100$ population size and $G=200$ generations) compared to LLEGO ($N=G=25$). Despite GATree's larger allocation (~$2$x runtime, ~$10$x # of function evaluations), LLEGO achieved superior performance, highlighting the effectiveness of semantic priors, fitness/diversity guidance, and higher arity in improving search efficiency.
>
> **Performance-computation tradeoff.** Our analysis demonstrates that LLEGO trades computational overhead for improved search efficiency and generalization performance. This trade-off is particularly advantageous in: (1) safety-critical domains where performance improvements directly impact outcomes (e.g. healthcare diagnostics, financial risk assessment), and (2) scenarios where function evaluation is expensive (e.g. physics simulations, hardware optimization). In these settings, LLEGO's ability to find superior solutions with fewer evaluations justifies its computational requirements.
>
> **Future enhancements.** A crucial direction for future works is to reduce computational requirements while maintaining performance. To do so, we recognize several promising directions: (1) reducing runtime through inference acceleration techniques such as speculative decoding and vLLM serving [1], and (2) reducing memory requirements through specialized fine-tuned models or quantization [2].
>
> **Actions taken:** We have included discussion of computational trade-offs and optimization strategies in `L483`.
>
> [1] Leviathan, Y., Kalman, M. and Matias, Y., 2023, July. Fast inference from transformers via speculative decoding. In International Conference on Machine Learning (pp. 19274-19286). PMLR.
>
> [2] Han, S., Mao, H. and Dally, W.J., 2015. Deep compression: Compressing deep neural networks with pruning, trained quantization and huffman coding. arXiv preprint arXiv:1510.00149.
>
> ---
>
> *We hope the reviewer’s concerns are addressed and they will consider updating their score. We welcome further discussions.*

---

> > ### Comment · Reviewer_71f5 · 2024-11-26
> > **Maintain my score**
> >
> > The authors have addressed my questions, I appreciate their efforts in clarifying the raised concerns. I am maintaining my original score.

---

> > > ### Author Response · Authors · 2024-11-28
> > > **Thank you**
> > >
> > > Dear Reviewer **71f5**,
> > >
> > > We sincerely appreciate your thorough engagement with our work and the concrete feedback that helped strengthen the paper. We are pleased that our responses addressed your comments and led to your recommendation for acceptance.
> > >
> > > With thanks,
> > >
> > > The Authors of #12082

---

### Official Review · Reviewer_v13n · 2024-11-03

**Soundness:** 4
**Presentation:** 4
**Contribution:** 3
**Rating:** 8
**Confidence:** 4

**Summary:**

This paper presents an application of LLMs as a crossover and mutation operator in an evolutionary algorithm. The authors propose “LLM-Enhanced Genetic Operators” (LLEGO), which uses fitness guided crossover and diversity guided mutation to infer decision trees from data. The proposed method (LLEGO) is compared against other decision tree induction algorithms on classification and regression tasks.

**Strengths:**

- The paper is very well written and easy to follow. The motivations for the proposed methods are detailed well.
- The experiments are well detailed, fitting, and the proposed approach is compared against a good number of baselines. There were very rigorous additional results, addressing issues such as fairness and generalization. Upon the first read, I was concerned that the results might be highly conditional on the solutions existing in the training set of the LLMs, but Appendix D.2 settled my concern.
- The use of LLMs in the loop of an evolutionary algorithm for decision tree induction is, to my knowledge, novel and, in my opinion, very promising.

**Weaknesses:**

- I would have been very interested in seeing a different use of selection algorithms, as fitness-proportionate (roulette wheel) selection is somewhat outdated and known to result in low population diversity. For example, perhaps the use of tournament or lexicase selection could adequately balance fitness and diversity?
- You use mutation as purely a technique to inject diversity into the population. I would have been interesting in seeing how “fitness-guided” mutation would perform (i.e. remove XO, and prompt the LLM to generate a better version) as a baseline. This would help motivate the use of XO/mutation as a means to explore/explore, respectively.

small typo:
- At the beginning of section 3 - “buid” should be “build”

**Questions:**

- The choice of using natural language as the representation for the evolutionary algorithm is fitting  as they need to be used by an LLM-based genetic operation. Do you imagine there being any ways to improve the syntactic validity from ~86% of the decision trees being outputted by your genetic operators?
- Is there any reason (beyond representation in natural language) that this method could not be used for symbolic regression or more general function induction?
- How well would the method work when starting without initializing with CART models on 25% of the training data?

---

> ### Author Response · Authors · 2024-11-21
> **Response to Reviewer v13n**
>
> *We appreciate the reviewer’s detailed and thoughtful evaluation and positive feedback.*
>
> ---
>
> ### [P1] Different parent sampling mechanisms
>
> Thank you for this concrete suggestion regarding selection mechanisms. We agree that LLEGO can accommodate different parent sampling schemes. In its current form, we utilized the roulette wheel mechanism, as the main focus of our crossover operator was the use of computed $f^*$ to guide offspring generation.
>
> To obtain insights into how LLEGO performs with different parent sampling approaches, we replace the roulette wheel selection with tournament selection, specifically varying tournament sizes $k \in $ {$1, 2, 3, 5$}, while fixing other aspects of the method. We included these additional results in `App E.5`. Our results reveal that larger tournament sizes yield higher population fitness but decreased diversity, due to increased selection pressure towards fitter individuals. Conversely, $k=1$ (equivalent to uniform sampling with replacement) maximizes diversity but reduces fitness pressure. Lastly, we observed that our current roulette wheel selection mechanism strikes an effective balance between these extremes.
>
> **Action taken**: In response to your feedback, we have included additional analysis on tournament selection in `App E.5`.
>
> ---
> ### [P2] "Fitness-guided" Mutation
>
> Thank you for this suggestion. As you correctly note, our dual operator approach intentionally separates exploitation (crossover) and exploration (mutation) to enhance search efficiency.
>
> **Empirical evidence.** We would like to point to a few pieces of empirical evidence that support the effectiveness of this design. (1) The ablation study (`Fig 7`) demonstrates that both operators working in concert achieve superior performance compared to either alone; (2) `Fig 5-6` show how $\alpha$ and $\tau$ provide precise control over exploration-exploitation, with $\alpha$ guiding search toward higher fitness regions while $\tau$ maintains diversity through less likely offspring.
>
> **Additional results.** Following your suggestion, we evaluated replacing our dual-operator approach with a single fitness-guided mutation using an "improve the solution"-type prompt (in `App E.2`). Results across 7 classification tasks (3 seeds) show LLEGO achieving consistently better performance, suggesting that separating exploration and exploitation roles between specialized operators is more effective than combining them.
>
> **Actions taken:** We have included additional results in `App E.2`.
>
> ---
>
> ### Questions
>
> * **Improving syntactic validity**: We appreciate this observant question. As you noticed, our implementation achieves ~$86$% validity rate when parsing LLM-generated offsprings back into machine-executable tree representations. Through manual error analysis, we found that the majority of invalid offspring were due to incorrect nesting structure (e.g. missing closing brackets) in our JSON-based representation. While this validity rate is sufficient for practical applications, potential improvements could be found using more structured formats or enhancing prompt instructions.
>
> * **Extension to more general function induction**: Thank you for this insightful suggestion. LLEGO's core methodology—using LLMs to provide semantic guidance for genetic operators—could indeed extend to other function induction problems like symbolic regression and program synthesis. In principle, any domain where solutions can be represented in natural language and meaningful semantic priors exist could benefit from our approach. However, each domain would require specific adaptations to handle unique challenges, such as different solution representations and domain-specific constraints. We elaborate on these potential extensions and implementation considerations in detail in `App E.8`.
>
> * **Other initialization**: Following your suggestion, we evaluated LLEGO's performance when initialized with weaker models (CART trees trained on only $2$ samples vs our original $25$% of training data). Results in `App E.6` show that while weaker initialization leads to slower convergence, LLEGO still achieves competitive final performance. This suggests that while good initialization enables faster discovery of high-quality solutions, our method remains robust to initialization quality.
>
> * **Typo**: Thank you, this has been fixed!
>
> [1] Brad L Miller, David E Goldberg, et al. Genetic algorithms, tournament selection, and the effects of noise. Complex systems, 9(3):193–212, 1995.
>
>
> ---
>
> *We hope the reviewer’s concerns are addressed and they will consider updating their score. We welcome further discussions.*

---

### Author Response · Authors · 2024-11-21
**Global Response**

We thank the reviewers for their constructive comments and their commitment to the review process.

We are encouraged that reviewers recognized LLEGO as both "novel" (**v13n**, **BF7z**) and a "creative and impactful idea" (**71f5**), representing a "significant advancement" in decision tree induction (**71f5**). We are glad that they found LLEGO's framework "well-motivated" (**BF7z**, **v13n**), with reviewers particularly appreciating that it was "easy to follow" (**BF7z**) and included a "detailed explanation of the LLEGO framework" (**71f5**).

On the experimental side, reviewers highlighted our "well detailed, fitting" experiments that included comparisons against "a good number of baselines" (**v13n**). The "promising results" (**mCct**) demonstrated "superior performance" (**71f5**, **BF7z**) and suggested LLEGO's potential to be "beneficial across various domains" (**71f5**).

We have also taken the reviewers’ feedback into account and made the following key improvements to the paper:
* Extended our empirical analysis with results on depth $5$ problems (`App E.1`)  and ablations on alternative prompting strategies (`App E.2`)
* Evaluated LLEGO's performance across different LLMs, specifically GPT-4 (`App E.4`)
* Investigated the impact of various parent sampling mechanisms (`App E.5`)
* Outlined potential extensions of LLEGO to other function induction domains (`App E.8`)

We have highlighted these changes in teal in the updated manuscript. We sincerely thank the reviewers for their valuable feedback on strengthening our work and remain open to further suggestions.


With thanks,

The Authors of #12082

---

### Meta-Review · Area_Chair_PkR4 · 2024-12-19

**Metareview:**

Summary:
This paper presents LLEGO, a novel framework that integrates LLMs with GP to enhance decision tree induction. LLEGO incorporates semantic priors and domain knowledge into the optimization process via LLM-based fitness-guided crossover and diversity-guided mutation operators. By representing decision trees in natural language, the framework allows for broader contextual understanding and higher-arity operations. Empirical results on diverse benchmarks highlight LLEGO's superior performance in terms of search efficiency and the quality of the evolved decision trees compared to existing tree induction methods.

Strengths:
(1) The combination of LLMs and evolutionary algorithms for decision tree induction looks innovative. (2) Empirical results demonstrate significant improvements over conventional GP methods. (3) The paper is well-written, easy to understand, and includes effective visualizations.

Weaknesses:
The motivation for using LLMs in decision tree induction is not sufficiently justified, which makes it unclear why LLMs are particularly beneficial for this task.

Decision:
The strengths of this paper outweigh its weaknesses, and I recommend acceptance. However, the authors are encouraged to address the reviewer feedback by clarifying how the LLM functions as an implicit regularization mechanism to mitigate overfitting in the final version of the paper.

**Additional Comments On Reviewer Discussion:**

Three out of four reviewers have given positive ratings for this paper. The remaining reviewer, while maintaining a score of 5, does not object to its acceptance. This reviewer acknowledges the strengths of the work but highlights a weakness regarding the lack of clarity on how the LLM functions as an implicit regularization mechanism to mitigate overfitting.

In my opinion, the authors have addressed most of the reviewers' concerns, and the paper meets the standards of ICLR. However, it is essential for the authors to carefully address the final concern raised by this reviewer in the final version.

---

### Decision · Program_Chairs · 2025-01-22

Accept (Poster)